# Implantable bioelectronics for gut electrophysiology

Alexander J. Boys [1,2,15], Amparo Güemes [3,15], Liang Ma[4], Rohit A. Gupta [5], Zixuan Lu [1], Chaeyeon Lee [3], Salim El-Hadwe [3,6], Alejandro Carnicer-Lombarte [3], Tobias E. Naegele [3], Friederike Uhlig [7,8,9], Damiano G. Barone [3,6,10,11], David C. Bulmer[5], Jennifer N. Gelinas [12,13,14], Niall P. Hyland [7,8], Dion Khodagholy[4], George G. Malliaras [3] ✉ & Róisín M. Owens [1] ✉

A major regulator of gastrointestinal physiology is the enteric nervous system. This division of the autonomic nervous system is unique in its extensiveness, with neurons distributed from the esophagus to the rectum, and its capability for local information processing. However, the constant motion of the gut, arising from its relative movements in the peritoneal cavity and the peristaltic movements associated with gut motility, as well as the sparse distribution of the neurons constituting the enteric nervous system, has made access and analysis exceedingly challenging. Here, we present the construction and validation of a bioelectronic implant for accessing neural information from the distal colon. Our bioelectronic monitoring system demonstrates real-time electrophysiological recording in response to chemical and mechanical distension under anesthesia and to feeding and stress in freely-moving animals. Direct access to the communication pathways of the enteric nervous system paves the way for neuromodulation strategies targeting the gut–brain axis.

The gastrointestinal (GI) tract contains large networks of electrically active cells that drive gut function through their interplay. Among these networks is the enteric nervous system (ENS), a network of sensory, inter- and motor neurons that plays a major role in controlling gut sensory transduction and motility, ion and mucus secretion, and vasodilation[1,2]. The ENS is also an essential part of the bidirectional gut-brain axis, affecting health and behavior[3,4], with many newly discovered connections to behavioral and pathological changes elsewhere in the body[4–7]. Unlike other areas of the peripheral nervous system, the ENS is also capable of processing local information through reflex-like circuits that act independently of the central nervous system (CNS). For example, it can coordinate simple peristaltic function independent of the central nervous system[1]. In addition to the ENS, the GI tract also contains electroactive pacemaker cells, called interstitial cells of Cajal (ICCs), that guide slow-wave gut motility, as well as a layered series of smooth muscle. The prominence of the gut-brain axis in human health, the relative independence of the ENS from the CNS and the possible role of the ENS as a relay for gut-to-brain signaling has made exploration of local gut electrophysiology timely.

[1]Department of Chemical Engineering & Biotechnology, University of Cambridge, Cambridge, UK. [2]Thayer School of Engineering, Dartmouth College, Hanover, NH, USA. [3]Department of Engineering, University of Cambridge, Cambridge, UK. [4]Department of Electrical Engineering, University of California, Irvine, CA, USA. [5]Department of Pharmacology, University of Cambridge, Cambridge, UK. [6]Department of Clinical Neurosciences, University of Cambridge, Cambridge, UK. [7]Department of Physiology, University College Cork, Cork, Ireland. [8]APC Microbiome Ireland, University College Cork, Cork, Ireland. [9]Department of Pharmacology and Therapeutics, University College Cork, Cork, Ireland. [10]Department of Neurosurgery, Houston Methodist Hospital, Houston, Texas, USA. [11]Neuroengineering Initiative, Rice University, Houston, Texas, USA. [12]Department of Anatomy and Neurobiology, University of California, Irvine, CA, USA. [13]Department of Pediatrics, University of California, Irvine, CA, USA. [14]Children's Hospital of Orange County, Orange, CA, USA. [15]These authors contributed equally: Alexander J. Boys, Amparo Güemes. ✉e-mail: gm603@cam.ac.uk; rmo37@cam.ac.uk

The ENS is organized in two interconnected ganglionated plexi (myenteric and submucosal) that run along the length of and wrap circumferentially around the GI tract[1], with the former plexus situated between the outer muscle layers and the latter plexus in the submucosa. Given their positioning within the walls of the gut, these plexi are difficult to access in live animals. While genetically encoded calcium indicators, like GCaMP, enable non-invasive imaging of neuronal activity, their reliance on genetic modification limits their applicability in certain studies[8]. This issue, coupled with the limited and low-amplitude signaling from the electroactive cells in the gut, adds further challenge to monitoring the complex interplay of electrophysiological signaling within the gut. Traditional electrophysiological recording devices, like metal penetrating microelectrode arrays and silicon surface electrodes, are not compatible with the constant movements of the gut and fail to maintain conformal contact with the highly contoured and elastic tissue that constitutes the GI tract. Consequently, previous studies have predominantly relied on ex vivo recordings in organ baths, employing intracellular techniques, most commonly using sharp electrodes[9] or voltage-sensitive dye (VSD) imaging[10]. Calcium imaging has also been used in vivo under anesthesia to indirectly assess neural activity and synaptic dynamics[11,12]. Recent advances in the development of flexible bioelectronic probes have allowed for the construction of devices that are compatible with soft tissues and maintain a sustained electrical interface with local cell populations[13–15]. These devices function in an extracellular recording capacity, meaning that they can collect neural signals along with other electrophysiological inputs. Even with these advancements, few studies to date have been able to access intestinal tissue in situ using in vivo implantable technologies[16,17], and few have also been able to capture gastric electrical activity, such as slow waves, in anesthetized animals and humans[18–21]. However, recording high-resolution, putative single-neuron activity from the ENS in the gut of freely moving animals remains an unmet technical and biological challenge.

Here, we develop a custom and conformable bioelectronic implant system for conducting in vivo electrophysiological recordings within the colonic wall of rodents. First, we record local gut electrophysiology elicited in response to various in vivo mechanical and pharmacological stimuli in anesthetized rodents. We then present an updated set of implants with integrated backend electronics and an associated surgical technique for performing recordings in freely moving rats. This setup allowed us to examine the physiological electrical activity of the ENS and the colonic response to food intake and stress, discerning changes in electrophysiological response to both stimuli. We additionally show the scalability of these devices across multiple species, indicating their utility as a broad neurogastroenterological tool. This study opens up the potential to monitor ENS activity continuously in real time along the length of the GI tract, to not only increase our fundamental understanding of the ENS and GI physiology, but also to understand the gut-brain axis' influence on behavior.

## Results and discussion
### Implant design & surgery development
We designed an implantable bioelectronic device and associated surgery for recording electrophysiological signals from the gut in vivo (Fig. 1a, b). The devices are based on state-of-the-art bioelectronic photolithography microfabrication principles[14,22,23], using a flexible, dielectric substrate for tissue contact (parylene-C) and a series of open recording gold electrodes coated with the conducting polymer, poly(ethylene dioxythiophene):poly(styrene sulfonate) (PEDOT:PSS), reducing impedance and therefore electrode size, for improved recording capabilities[24], which is visible via electrical impedance spectroscopy (Supplemental Fig. 1). The devices were designed to meet the required size specifications to reside within the walls of the colon (see details in Methods) and a tetrode layout (Fig. 1c) to resolve

the neuronal clusters of the ganglionated plexi of the ENS[1]. We additionally included a series of markers and loops to assist in the surgical placement of the device in juxtaposition with the ENS.

For surgical access, we performed a laparotomy and isolated the colon from the surrounding tissue (Fig. 1d, Supplemental Fig. 2). To place the device, we ran a needle underneath the *muscularis externa* of the colon (Fig. 1e) and then back-tracked along this tunnel using a pair of reverse-action forceps (Fig. 1f). We then located and gripped the leading edge of the implant (Fig. 1g) and threaded the implant through the tunnel with the electrodes facing luminally to record from the submucosal plexus (Fig. 1h, i). The placement of the implant within the colonic wall also facilitates tight contact with tissue, even during acute and chronic recordings, as demonstrated below. The flexible device substrate and the inherent elasticity of the gut tissue also limit the effect of blood, other accumulating fluids, and breathing interference on the quality of the electrophysiological recordings (Supplementary Movies 1, 2). To confirm appropriate placement of the implant, we performed histology for device placement (Fig. 1e) into the colonic wall, identifying the positioning of the tunnel as luminally adjacent to the *muscularis externa* without perforating the gut wall or damaging the submucosal layers (Fig. 1j, Supplemental Fig. 3l). To reduce interference from the myenteric plexus, we placed the insulating backing of the implant towards the *muscularis externa*, thereby also facing the electrodes towards the submucosal plexus of the ENS, which limits electromyographic signal contribution.

### Distension of gut wall validates physiological responses to mechanical stimuli
During the passage of fecal matter through the colon, the gut wall experiences substantial strain, which activates stretch-sensitive neurons and drives a cascading peristaltic response[2,25,26]. To assess our device's performance, we initially analyzed the electrophysiological response of the colon to distension, validating the device's monitoring capabilities in vivo under anesthesia. To initiate a distension, we ligated ~1 cm of colon to fluidically isolate this segment. We implanted the monitoring device as described above (Fig. 1d–i) and then, we placed a needle into the lumen of the colon in the vicinity of the implant (Fig. 2a) and injected ~0.3 mL of saline using a syringe pump, causing an increase in intraluminal pressure and distending the tissue (Fig. 2b, Supplementary Movie 3), while recording the evoked electrophysiological activity.

We performed two such distensions under low anesthetic dose (1.3% isoflurane), then increased the anesthetic dose (5% isoflurane), and performed two further distensions. We examined the electrophysiological response at both high frequency (300–2000 Hz), to isolate the neural components from the signal, and low frequency (0–300 Hz), to isolate the slower frequency components of the signals arising from other electrically active cells in the network, like submucosal and myenteric ICCs and smooth muscle cells (Fig. 2c), in 10 s windows after each distension (Fig. 2c). We observed an initial fast peak in the high-frequency trace (Fig. 2d), followed by an asynchronous extended voltage response, visible in the low-frequency trace (Fig. 2e), before returning to baseline. At the higher isoflurane dose (5%), neither high nor low frequency responses were observed (Fig. 2f, Supplemental Fig. 4a–f). This 'silencing' response may be explained by a direct reduction of neural activity[27,28] or by an anesthetic-induced muscle relaxation[29], leading to an elevated threshold for eliciting a response. In other words, the muscle's state of relaxation under the influence of higher doses of isoflurane may require a higher stimulus or pressure to activate neural responses. To examine the specific effects of the isoflurane anesthetic, we also performed recordings using urethane, which is known to have limited neural suppression[27]. We observed a more consistent response (Supplemental Fig. 4g, h), but we found that the typical route of application for urethane through an intraperitoneal injection was infeasible for repeat experiments, given

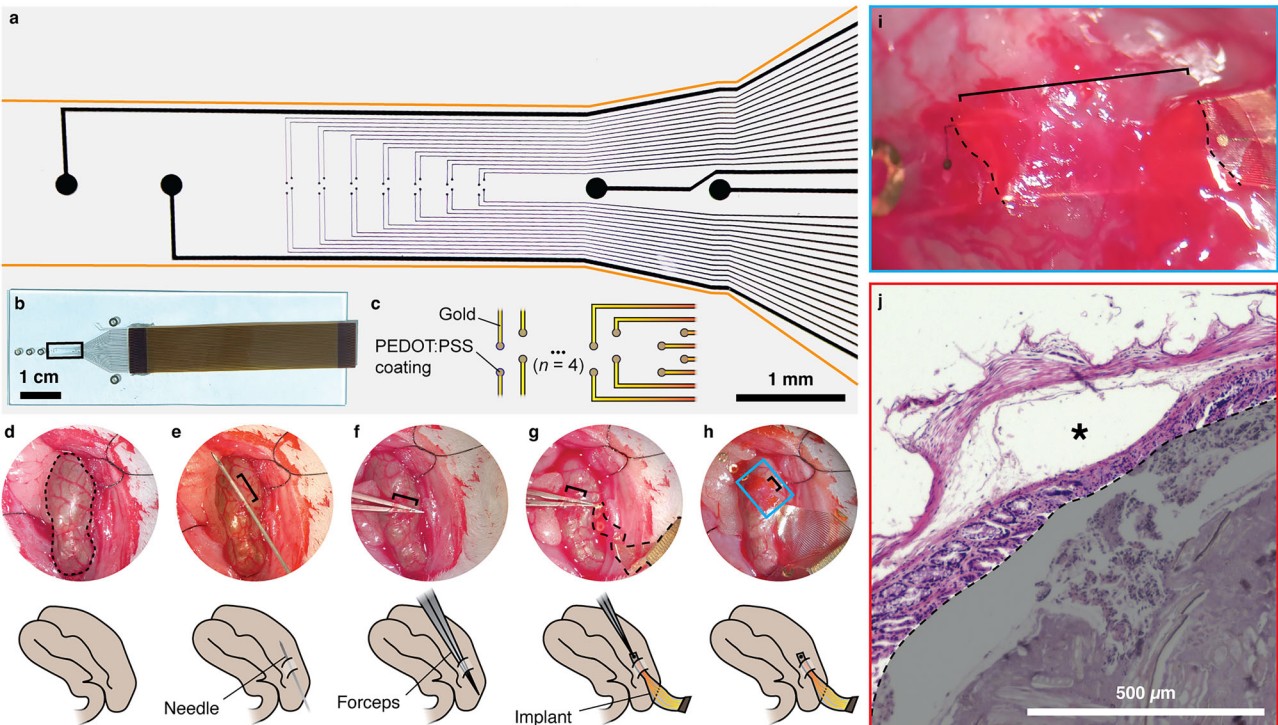

**Fig. 1 | Conformable devices can be surgically implanted on top of the sub-mucosal plexus of the colon. a** Brightfield image taken in transmission showing the device layout for accessing the ENS. Devices consist of gold tracks insulated with parylene-C with poly(ethylene dioxythiophene):poly(styrene sulfonate) (PEDOT:PSS)-coated gold electrodes, containing a central array of 28 recording electrodes, each 20 μm in diameter, flanked on either side by a pair of large electrodes, 200 μm in size, which are wired but were used as surgical markers. Given that the device substrate (parylene-C) is translucent, the outline of the device has been highlighted in orange. **b** Photograph of device with bonded flat flexible cable (FFC) for connection to Intan recording system. Device is situated on a glass slide in this image for transport, but is removed for implantation. Black box indicates the region of device shown in a. Loops designed into the device for suturing are visible on the left, outside the black box. **c** Schematic detailing recording electrode layout, where electrodes are arranged as a linear array of 7 tetrodes, with each electrode separated by 50 μm from its tetrode neighbors. **d**–**h**, Steps for implanting the

device into the colonic wall, images taken via stereoscope during surgery. Top shows surgical images of the steps, and the bottom shows a representative schematic for each image. Dotted line in d indicates the periphery of the section of colon under analysis. Brackets in e indicate the section of the needle that is situated in the wall of the colon. Brackets in (**f**, **g**) indicate the section of forceps that are within the wall. Dotted line in (**g**), highlights the outer edges of the implant. Brackets in (**h**, **i**) show the section of the implant that is in the colonic wall. Inset from (**h**) in the blue box is shown in (**i**) with the device situated beneath the *muscularis externa*. Dotted lines indicate where the device enters (bottom dotted line) and exits (top dotted line) the tissue. **j** Hematoxylin and eosin histological stain of needle placement in colonic wall, which is used to create the tunnel for the implant, as shown in (**e**). This image shows implantation directly below the *muscularis externa*. Asterisk indicates the tunnel in the tissue where the needle was threaded through the colonic wall. The lumen of the gut, which contains fecal matter, is shaded in gray.

the location of the colon within the peritoneal cavity. For isoflurane recordings, this two-component signal, featuring an initial fast peak in the high-frequency trace followed by a slower, extended low-frequency response, suggests the involvement of multiple cell types. The high-frequency activity reflects neuronal firing, consistent with the primary role of the ENS in initiating the colonic motor response, and confirms the close proximity of the electrodes to submucosal neurons. High-frequency signals from more distant sources, such as the myenteric plexus or muscle layers, are likely attenuated due to the low-pass filtering properties of biological tissue and the dielectric backing facing the myenteric plexus and smooth muscle layers[30]. In contrast, the slower low-frequency component likely arises from the integrated activity of the broader neuromuscular network. Although the device is oriented away from the myenteric plexus, low-frequency signals from more distant sources, including smooth muscle and ICCs in both the submucosal[31] and myenteric layers, which are electrically active as demonstrated in the gastric motility network[32], can still propagate through tissue and be detected. We note that while colonic motility is predominantly neurally mediated, there is evidence that myogenic mechanisms can contribute under certain conditions[33]. We leveraged this physiological feature to validate device placement and recording capabilities via distension-evoked responses. These experiments confirmed that recorded signals using this device setup are

electrophysiological in origin and that these devices are capable of accurate recording in the dynamic colonic environment.

## Pharmacological stimuli initiate multifrequency electrophysiological response with common neurophysiological characteristics

Following successful recordings of distension-associated electrophysiological responses, we conducted further validation of the recording capabilities of our devices by capturing elicited compound activity in response to sensory neural stimulants under anesthesia (Fig. 3, Supplemental Fig. 5). We recorded electrophysiological responses to the prototypic noxious inflammatory stimuli; bradykinin, an established chemical nociceptive stimulant relevant to tissue injury and pain which activates G-protein-coupled receptors[34], and activation by capsaicin, the active ingredient in peppers which activates transient receptor potential vanilloid type 1 (TRPV1) channels expressed on nociceptive sensory neurons[35]. Concentrations of 1 μM bradykinin, delivered topically (Fig. 3a), and 500 nM capsaicin, delivered topically (Fig. 3b) or intraluminally (Fig. 3c), were motivated by ex vivo electrophysiological recordings from lumbar splanchnic nerve (LSN) bundles in the mouse colon, which revealed colonic afferent responses to these mediators (Supplemental Fig. 5e, f). Drug additions were performed in sequence, beginning with bradykinin doses and followed

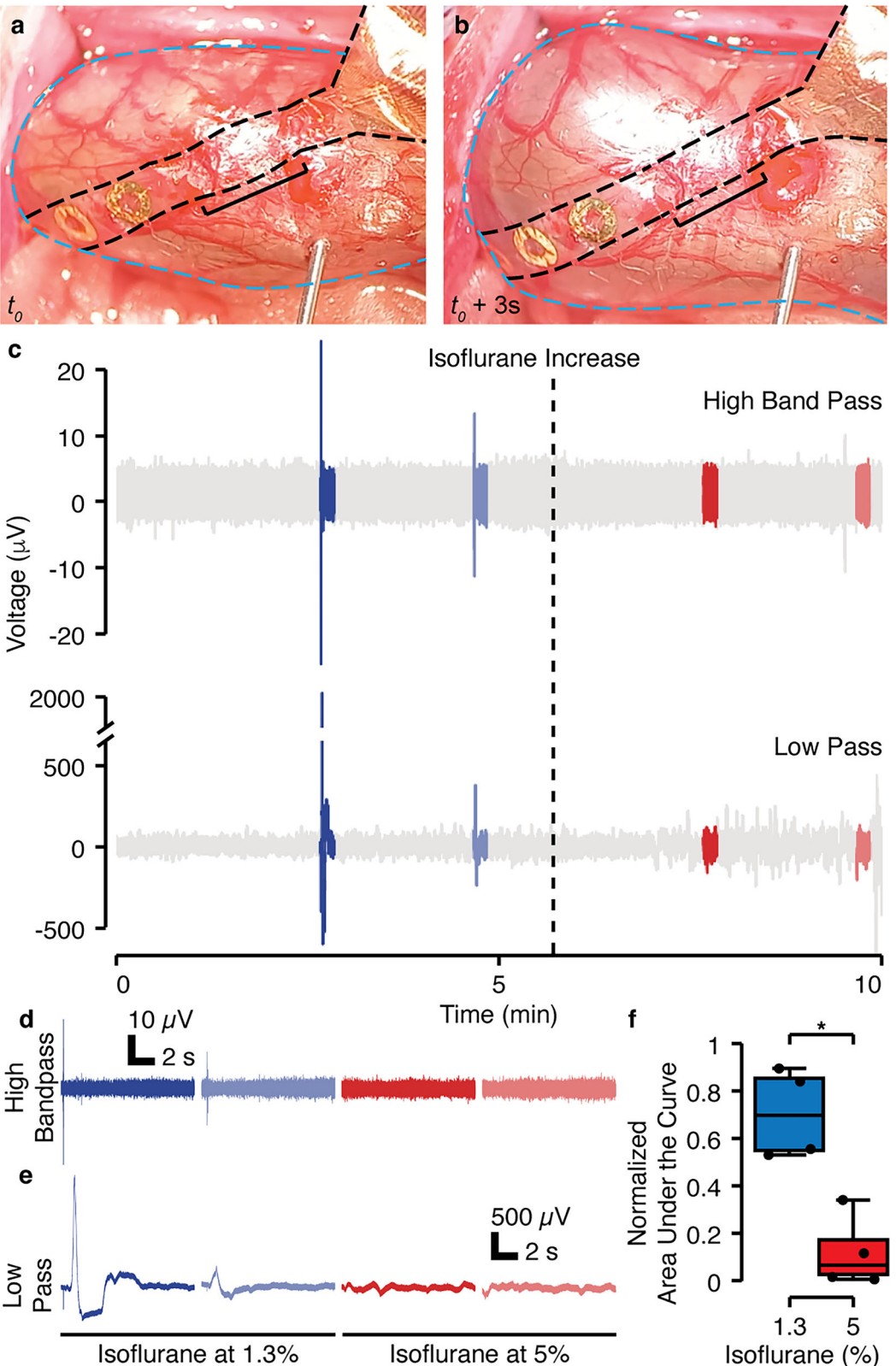

by capsaicin doses (Supplemental Fig. 5a–c), to account for the known desensitizing properties of capsaicin[36]. Each administration, unless specifically noted for an experimental test, e.g., desensitization, was followed by multiple washes with saline and a waiting period of at least 5 min to reduce cross-effects from different drugs. The intermediate washes also provide a non-pharmacological comparative signal, showing no response (Supplemental Fig. 5d).

Using this setup, we were able to record distinctive responses by examining the frequency-based power spectra (Fig. 3). Topical administration of bradykinin (Fig. 3a) occurred at $t_0$, followed by a brief period of inactivity (-10 s), then a burst of signal with a signal maximum around 1500 Hz at $t_1$, with a following low frequency increase at $t_2$, and return to baseline after -90 s. Topical capsaicin (Fig. 3b) showed a similar delay and response, suggesting sensory

**Fig. 2 | Conformable electronics enable in vivo acquisition of gut electro-physiological responses to in vivo mechanical stimuli. a** Ligated portion of the colon with a device implanted into the colonic wall prior to distension at $t_0$. **b** Distended colon during saline injection. For a and b, the colon is outlined with a dashed blue line. The implant is outlined with a dashed black line, and a bracket is used to designate the portion of the implant that is within the colon wall. **c** High-frequency bandpass (300–2000 Hz) and low pass (<300 Hz) representative full voltage traces. Segments highlighted in color represent the distension and the following 10 s intervals after distension. Isoflurane was initially at 1.3% (blue) and was increased to 5% (red) at the time represented by the dashed line. **d** Zoom in on the high-frequency bandpass traces corresponding to the distension segments. **e** Zoom in on the low-pass traces corresponding to the distension segments. **f** Normalized Area Under the Curve (nAUC) for all the low-frequency distension traces (10 s window) before and after the rise in isoflurane concentration ($n = 4$ rats,

2 values as low isoflurane and 2 values at high isoflurane per rat, see Supplemental Fig. 4a–f for traces from other rats). The nAUC standardizes the AUC obtained from each 10 s window by the range value within each experiment, ensuring a fair assessment of the responses across different rats. *: Statistically significant difference with $p = 0.003$ (t-statistic: 3.54, degrees of freedom: 14, Cohen's d: 1.77, 95% Confidence Interval of mean difference: [0.409, 0.763]) using two-sided unpaired t-test after criteria for normality and variance homogeneity assumptions were met. Statistics for boxplots: [blue (1.3 % isoflurane): minima (0.53), maxima (0.89), median (0.69), Q1 (0.54), Q3 (0.85), whisker low bound (0.09), whisker high bound (1.31), percentiles (0.53, 0.55, 0.69, 0.85, 0.88)], [red (5% isoflurane): minima (0.005), maxima (0.34), median (0.06), Q1 (0.01), Q3 (0.17), whisker low bound (−0.22), whisker high bound (0.41), percentiles (0.006, 0.01, 0.06, 0.17, 0.31)]. The quantification of the nAUC for the high-frequency segments was not deemed meaningful due to the short duration of the response (<20 ms).

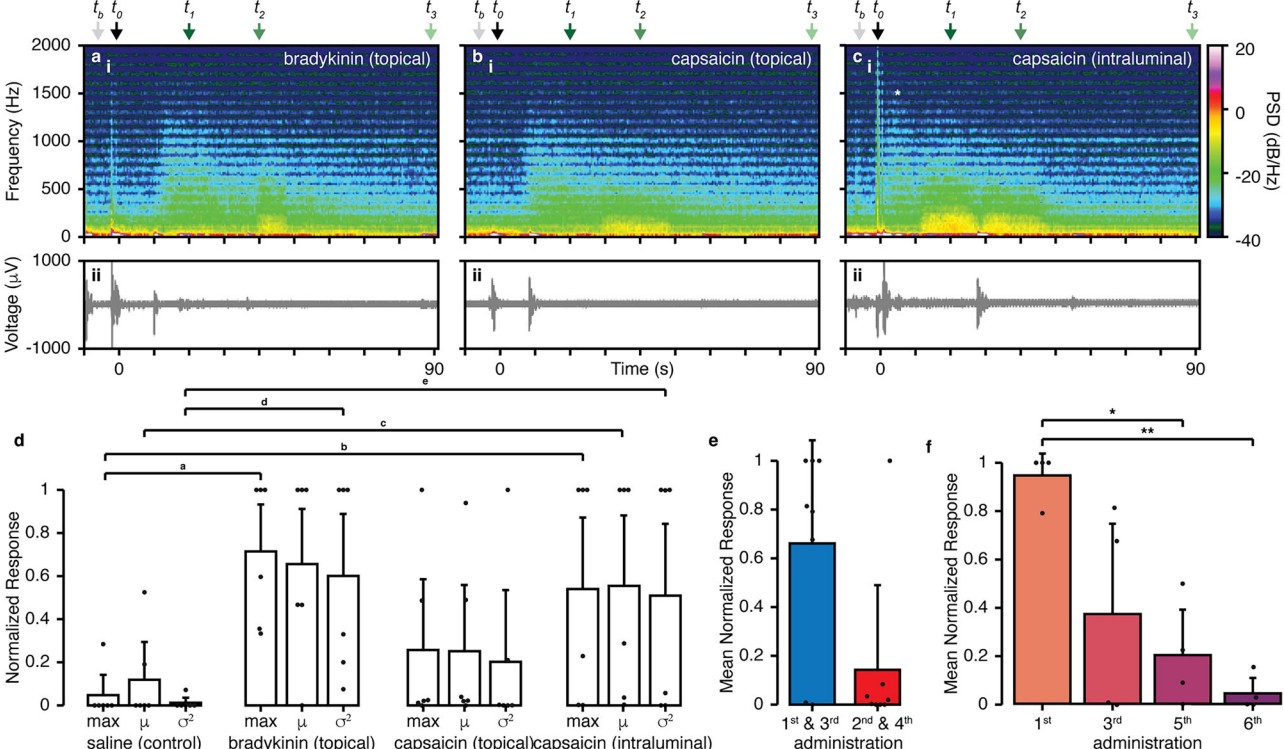

**Fig. 3 | Conformable electronics enable in vivo acquisition of distinct gut electrophysiological responses to various in vivo pharmacological stimuli.** Representative responses for (**a**) topical administration of 1 μM bradykinin, **b** topical administration of 500 nM capsaicin, and **c** intraluminal administration of 500 nM capsaicin (administered per the overall regimen shown in Supplemental Fig. 5), showing (i) spectrogram (0–2000 Hz) of power spectral density (PSD), (ii) raw voltage signal. Drug administration occurred at approximately $t_0$ for (**a**–**c**). Asterisk in c indicates initial high frequency response as discussed in the text. Power spectra corresponding to each data point, $t_b$, $t_0$, $t_1$, $t_2$, $t_3$ in the spectrograms for each drug are shown in Supplemental Fig. 6. **d** Normalized response per animal to different drug additions ($n = 6$ rats, independent experiments). The response activity elicited by each drug was quantified with the maximum [dB/(Hz*s)], mean [dB/(Hz*s)], and variance {[dB/(Hz*s)]²} of the temporal power spectral density (PSD), computed as the area under the curve (AUC) of the spectrogram power data using non-overlapping 1 s rolling windows. The responses were normalized [0,1] to the maximum and minimum values obtained among all additions (including saline) to allow for meaningful comparisons. Statistical summary: Mann-Whitney U Test: [a]$U = 0.0$, $p = 0.0034$, [b]$U = 3.0$, $p = 0.015$, [c]$U = 5.0$, $p = 0.04$, [d]$U = 0.0$, $p = 0.0034$,

[e]$U = 3.0$, $p = 0.016$; Bar plots overview: Max - saline (0.05 ± 0.12), bradykinin(0.71 ± 0.33), capsaicin topical (0.26 ± 0.41), capsaicin intraluminal (0.54 ± 0.51); μ - saline (0.12 ± 0.21), bradykinin(0.66 ± 0.41), capsaicin topical (0.25 ± 0.39), capsaicin intraluminal (0.55 ± 0.50); Var - saline (0.01 ± 0.03), bradykinin(0.60 ± 0.44), capsaicin topical (0.20 ± 0.40), capsaicin intraluminal (0.51 ± 0.54); e, e, f, Administration of bradykinin for (**e**) and (**f**) was performed as follows: 1st addition, 2nd addition, saline wash, 3rd addition, 4th addition, saline wash, 5th addition, saline wash, 6th addition. **e** Mean of the temporal PSD (300–2000 Hz) for an initial and secondary administration of bradykinin (repeated twice, separated 2 to 3 min in between−1st and 3rd additions compared to the 2nd and 4th) with no intermediate saline wash ($n = 4$ rats, independent experiments). Error bars: first (0.43), second (0.35). **f** Mean of the temporal PSD (300–2000 Hz) for 4 additions of bradykinin with at least one saline wash step in between ($n = 4$ rats, independent experiments). 1st addition compared to the 3rd, 5th, and 6th (Mann-Whitney U-test: *1st vs. 5th $U = 16.0$, $p = 0.0265$; **1st vs. 6th, $U = 16.0$, $p = 0.0265$; Error bars: first (0.09), second (0.37), third (0.19), fourth (0.06). Isoflurane was kept at 1.3% for the whole duration of the recordings.

activation. Intraluminal capsaicin (Fig. 3c) resulted in a stronger response. In this case, the delay period shows some high-frequency activity before a broad-frequency response, likely due to differences in drug absorption versus topical as well as the change in administration

route, before returning to baseline after 60 s. The initial transient high-frequency response is possibly linked to afferent sensory activation in the myenteric plexus and / or *muscularis externa*[37], complemented with the activation of TRPV1 receptors on submucosal neuroendocrine

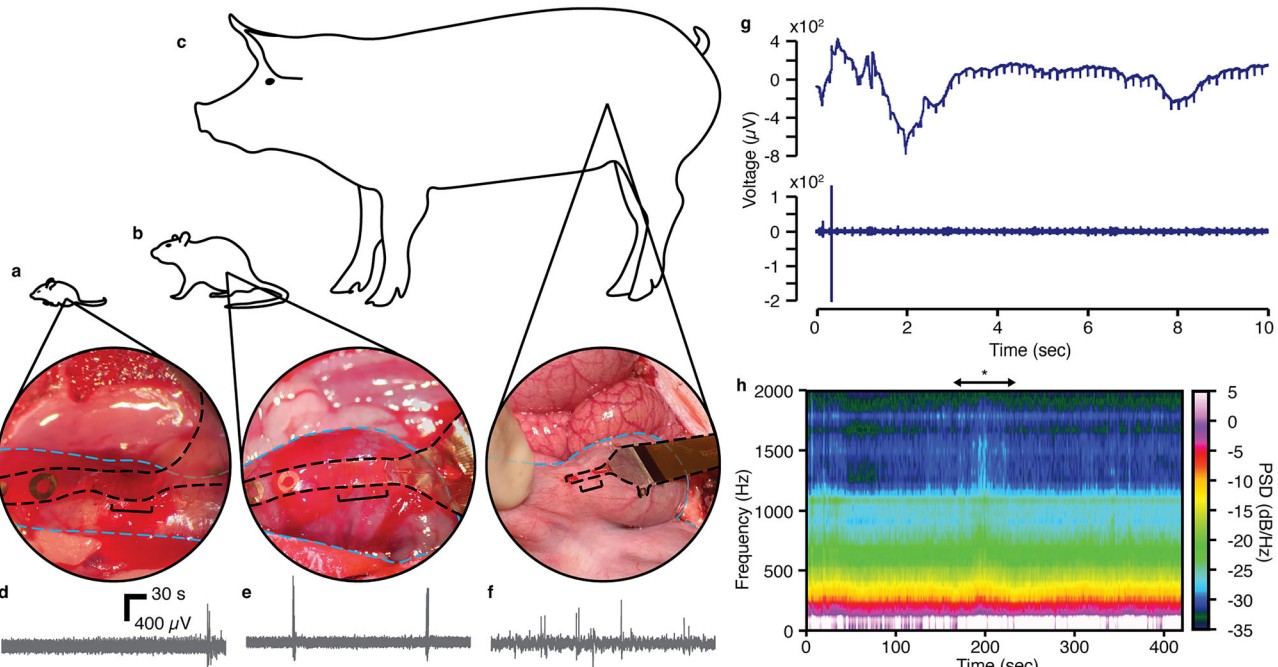

**Fig. 4 | Surgical placement and electrophysiological recording demonstration in mouse, rat, and pig.** Image showing surgical placement of implantable device in **a** mouse colon, **b** rat colon, and **c** pig colon. For each image, the device is outlined with black dotted lines, and the colon is outlined with blue dotted lines. The black bracket indicates the portion of the device that resides within the colonic wall. Representative voltage traces, bandpass filtered between 0 and 2000 Hz, for **d** mouse, **e** rat, and **f** pig. We see similar signal-to-noise ratios for each system. **g** Distension trace for mouse showing time-synced (top) low-pass (0–300 Hz) and (bottom) high-band pass (300–2000 Hz). **h** Spectrogram showing response to capsaicin administration in the pig colon. For this experiment, ~10 mL of capsaicin was poured topically onto the region containing the implant over the course of 1-min. This spectrogram begins (time at 0 s) at the culmination of the capsaicin dosing. The maximal visible response is highlighted using an asterisk. These pilot experiments were performed in a similar fashion to the rat data, although further refinement may be necessary if conducting an entire study in a different species. However, given the general conservation in neuronal density across vertebrate species[48,69], the data we collect should be of similar origin to those data from the rat.

cells when injected from the luminal side[38], which then triggers the transmission of electrical signals between electrically coupled muscle cells through gap junctions, resulting in sustained high-firing activity. Changes are evident in the individual power spectra at different data points (Supplemental Fig. 6).

The population results were quantified by normalized metrics extracted from the temporal power spectral density (Fig. 3d) and compared to saline additions (Supplemental Fig. 5d), used as control and always registering the minimum response. The addition of bradykinin and intraluminal capsaicin resulted in a consistent, substantial response, while the response to topical capsaicin varied in magnitude across animals. In all cases, some variability was observed between animals and administrations. This variability is likely due to challenges in precisely positioning the device within the colon wall, which affects proximity to neurons and electrophysiologically active cells. Furthermore, while numerous capsaicin-sensitive cells exist in the colon[39], not all neurons are capsaicin-sensitive[40], contributing to the heterogeneous responses observed between animals.

To examine whether repeated bradykinin additions induced measurable desensitization[41], using the same setup, we added bradykinin multiple times to the same tissue, with or without a saline wash between additions. To perform this experiment, we added bradykinin topically to the tissue six times in total with the following scheme: bradykinin (1st), bradykinin (2nd), wash, bradykinin (3rd), bradykinin (4th), wash, bradykinin (5th), wash, bradykinin (6th). To analyze these data, we compared the first and third additions with the second and fourth additions (Fig. 3e), finding that back-to-back additions of bradykinin result in a trending decrease in signal intensity for the second response. We also compared the first, third, fifth, and sixth additions (Fig. 3f), finding that the response decreases after each, with a significant decrease observed for the first addition compared to the fifth and to the sixth additions. These data indicate a general desensitization of the neural response to bradykinin, as would be expected[41].

### Extension of electrophysiological monitoring system to broader neuro-gastroenterological instances

Anesthesia generally results in reduced spontaneous gut activity. However, during these and later procedures, we noted a few instances of visible spontaneous contractions that likely occurred in response to previous mechanical perturbation during surgery. We successfully recorded the electrophysiology response of one of these contractions (Supplemental Fig. 7a, b), which was characterized by a large, broad-spectrum increase in electrophysiological power, which decreased back to baseline at the end of the visible contraction, approximately 1.5 min after initiation. The initial frequency maximum was approximately 1500 Hz, indicating a neuronally-driven response, followed by a lower maximum and increased signal intensity in sub-200 Hz frequencies (Supplemental Fig. 7b). A clustering analysis of identified waveforms also showed particular waveforms which were only present in the signal during this contraction (Supplemental Fig. 7a, d).

Finally, to showcase the broad usage and scalability of this device, we demonstrated its utility in recording similar electrophysiological data in both mouse and pig colon under anesthesia (Fig. 4, Supplemental Fig. 8). We showed that we could perform similar tests in both larger and smaller species, achieving similar data types, indicating the extension of this platform as a scalable neuro-gastroenterological tool. Overall, the multifaceted nature of these signals underscores the complexity of neural interactions within the network. When evaluating the resultant signal in all cases, we observe distinct patterns of activation observed with different routes of drug administration,

indicating the presence of various mechanisms of action, each exhibiting unique dynamics, especially in the high-frequency components of the signals (neural activation). These observations, along with our results from mechanical distensions, confirm our capability for recording gut electrophysiology in vivo.

## Chronic implantation of electrophysiological recording device enables high spatiotemporal probing of putative single neurons

Building on our successful validation of neural gut electrophysiology recordings in vivo under anesthesia, we shifted our focus to overcoming the limitations of recordings under anesthesia, particularly the strong suppression of spontaneous neural activity and the relative inactivity of the ENS under general anesthesia[7]. To truly understand the ENS within a fully functional gastrointestinal tract, we explored chronic recordings in freely moving animals. This approach allowed us to capture natural, spontaneous neural activity without the need for mechanical or chemical stimulation—an achievement not previously realized in unrestrained conditions.

To achieve chronic implantation of our devices, we modified the backend electronics while maintaining the same tissue interface format used for acute recordings (Fig. 5a, b). In this manner, electrode contact with tissue was preserved despite the substantial motion of the colon and surrounding tissues in a freely moving rat. Data were extracted through a percutaneous shoulder connector that permitted tethering to an acquisition system during recording sessions. The recording ground electrode was situated within the subcutaneous tissue adjacent to the percutaneous port (Fig. 5c). We developed this approach in a pilot study ($n = 2$ rats), demonstrating feasibility for recording electrophysiological activity in freely moving animals for up to 2 weeks (Supplemental Fig. 9a–d). We also showed the tissue response to the placement of this device (Supplemental Fig. 9e, f).

Using this experimental setup, we monitored electrophysiologic activity across multiple 30 min recording sessions ($n = 4$ rats, 3 recording sessions up to 12 days after implantation) to analyze specific neurophysiologic data and trends. Animal weight (Supplemental Fig. 10a) and implant impedance (Supplemental Fig. 10b–e) were monitored throughout the experimental period. Rats recovered remarkably well from surgery with no noted negative effects. Maintaining close anatomical proximity of the miniaturized, high-efficiency recording electrodes to the submucosal plexus, through the use of the surgical technique described above (Fig. 1), opened the possibility of detecting neural spiking activity from the ENS. We also employed a tetrode-based electrode configuration (Figs. 1, 4d), which is a geometric arrangement that facilitates differentiation of action potential waveforms from individual neurons. High-pass filtering (>300 Hz) revealed bursts of high-frequency activity that were restricted to nearby electrodes, suggestive of neural spiking (Fig. 5e). To investigate putative activity patterns attributable to individual gut neurons, we adapted a thresholding and clustering approach commonly used with brain-derived electrophysiologic recordings[24,42,43]. We identified 89 putative single neurons across all recording sessions. These neurons displayed anatomically plausible, localized spatial patterns of action potential waveforms across recording electrodes (Fig. 5f). Importantly, they also exhibited physiological spiking refractory periods (Fig. 5g), distinct patterns of co-activation with other neurons (Fig. 5h), and separable clusters of spatiotemporal action potential features in principal component space (Fig. 5i), supporting the characterization of this activity as representing individual neurons. Spikes from individual neurons were also detected across the time course of the session (Fig. 5j), indicating a stable interface between our device and the tissue.

## Conformable devices enable to record colonic responses to feeding and stress

The rats from the above electrophysiological analysis ($n = 4$) were used for behavioral recordings, where we recorded colonic electrophysiology in response to two key conditions: (1) acute stress induced by exposure to a novel environment for the first 15 min, followed by (2) the physiological response to food stimulus. At the 15-min mark, Nutella® was introduced into the same environment (Supplementary Movies 4, 5), allowing the rats to feed freely while recordings continued for at least another 15 min. Data were collected in the same manner on Days 1, 8, and 12.

We first analyzed the neural spiking patterns of putative individual gut neurons in response to feeding. We aligned the normalized instantaneous firing rate of each neuron to the onset of feeding and observed that most of the population exhibited an increase in activity within 2 min of this timepoint (Fig. 6a). Averaging the individual neuron responses to obtain a population measure revealed a significant enhancement of spiking in the post-meal compared to the pre-meal and meal epochs (Fig. 6b). A statistical comparison of putative single neuron average firing rate during pre-meal, meal, and post-meal epochs across 3 days of recording was found to be significant (Fig. 6c). These findings are in agreement with prior work indicating that enteric neural activity increases with food intake[44,45] and further support the ability of our device to capture physiologically relevant electrophysiologic signals from individual gut neurons.

Taking a broader, multi-cell approach, we investigated the gut response to a novel environment to assess anxiety-like behavior[46]. Given the expected widespread activation of diverse electrophysiological units present in the colon, including neurons, ICCs, and smooth muscle cells, we used a frequency band decomposition to capture and differentiate their distinct electrophysiological dynamics beyond isolated neural signals. We found that on Day 1 after implantation, the first time the animals were exposed to the novel environment (Fig. 6d), an initially large electrophysiological power was evident across frequency bands, especially within the first 5 min (Fig. 6e, Supplemental Fig. 11). This power trended downwards per minute as the rats spent further time in the environment. This electrophysiological behavior becomes less evident on Day 8 and Day 12 as the rats were re-exposed to the same novel environment (Supplemental Fig. 11). This trend is notable when compiling the overall power for each frequency band across the first 15 min in the novel environment per day (Fig. 6f). We propose stress acclimatization[47] as a mechanism accounting for these observed data. Notably, different frequency bands temporally varied independently over time, indicating the biologically dense electrophysiological data collected from the colon with our devices (Fig. 5e, Supplemental Fig. 12).

In conclusion, we present a strategy and associated device for achieving neural recordings from the ENS in anesthetized and freely moving animals. Our data shows that rational design of bioelectronic devices coupled with surgical innovation allows access to traditionally difficult-to-reach areas of physiology. The ENS presents a particularly challenging region of the nervous system to access, given its inherent susceptibility to suppression from anesthetic agents, its location within a highly mobile and elastic tissue, and its disperse and somewhat sparse distribution (ganglionated plexi of nerves) within the walls of the GI tract. Using current methodologies for flexible device fabrication, we were able to overcome these issues to record neural activity as well as other electrophysiological activity from the ENS. We further demonstrated the extended applicability of our device by successfully recording electrophysiological data from both mouse and pig colon, highlighting its versatility across different species. We show the successful recording of putative single neurons in freely moving animals, as well as the overall electrophysiological response of cells residing in colonic tissue to nutritional intake, as well as stress. These studies set the groundwork for exploring an important and consequential region of the nervous system and lay the framework for the democratization of neural recording technologies to the vast areas of the body that are supported by the peripheral nervous system.

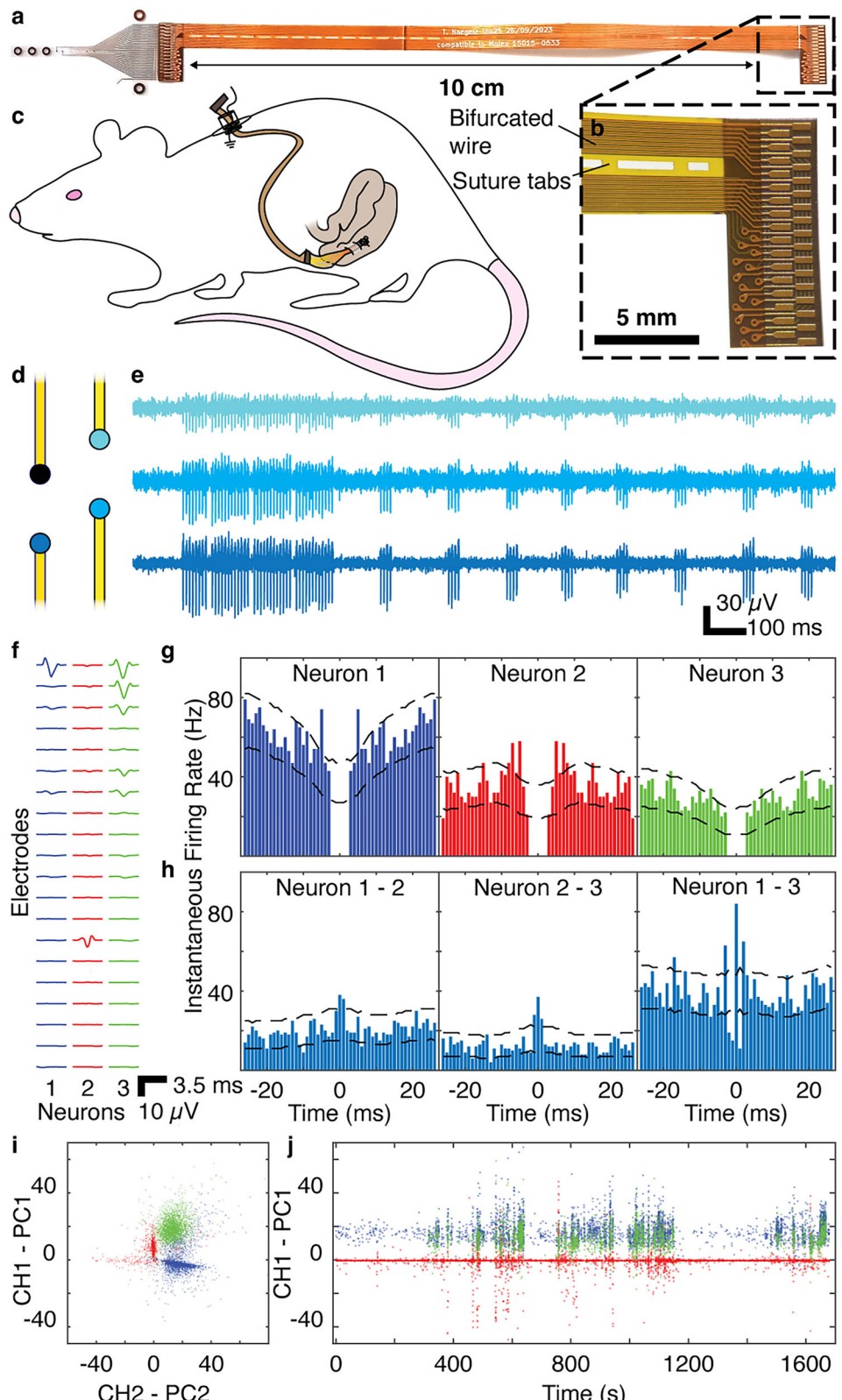

## Methods

### Device design

We designed our device to maximize the likelihood of recording from the dispersed ganglia of the ENS. Unlike other areas of the nervous system, the ENS consists of a net-like configuration of interconnected ganglia. The inter-ganglionic spacing in the myenteric plexus for proportions of the ENS in a rat is approximately 250 μm[48]. The submucosal plexus is more sparsely populated, resulting in larger distances between ganglia, but still within the same order of magnitude. This supports the relevance of our electrode design for capturing activity across both plexi. Given the inherent low-pass filtering properties of tissue[49], electrodes must be positioned in the vicinity of any high-frequency sources of interest in order to record these data. In the case of PEDOT:PSS-coated, gold

**Fig. 5 | High spatiotemporal resolution electrophysiologic acquisition permits identification of putative single neurons in the enteric nervous system.**
**a** Custom implantable wire was designed to facilitate chronic device implantation. The wire has two symmetric plugs / connectors, used to insert into a zero-insertion force (ZIF)-type plug (right), which, via a custom-printed circuit board (PCB), can be interfaced with an Intan headstage for recording. The other end of the wire has an identical geometry but is bonded onto the microfabricated implant (left), which has the same replicated connector on the device itself. **b** Inset showing plug with intermediate wiring visible on the left of the image. The intermediate wiring is double-sided to reduce footprint and bifurcated to allow for the placement of sutures around the wire. Small tabs connect the bifurcated wires across the length of the wire, enabling the wire to be anchored at the point where it crosses the peritoneal wall for access to the colon. **c** Schematic detailing the placement of wiring and device within the rat. For the surgery, the wiring was routed subcutaneously between the percutaneous port down to the ventrolateral flank of the rat, where it crossed the peritoneal wall, allowing for implantation of the microfabricated portion of the device into the colon. **d** Illustration of the tetrode

arrangement of four electrodes in one device in a chronically implanted rat.
**e** Sample high-pass filtered traces obtained during a recording session conducted 12 days post-surgery from three implanted electrodes (color refers to matching electrode in (**e**) during (right; scale bar, 100 ms and 30 μV). **f** Averaged extracellular spike waveforms for three sample putative single neurons recorded from the implanted electrodes presenting acceptable (<500 kΩ) electrical impedance, which also presented good signal quality confirmed through offline analysis in the same chronic recording session demonstrating distinct and anatomically consistent localization across electrodes (*n* = 4718 spikes/neuron 1, 4748 spikes/neuron 2, 2497 spikes/neuron 3; trace, 3.5 ms, right; scale bar 3.5 ms and 10 μV).
**g** Autocorrelations of spike occurrence for putative single neurons shown in b demonstrate physiologic refractory periods. **h** Cross-correlations of spike occurrence between putative single neurons reveal different co-activation patterns.
**i** Spatiotemporal features of putative single neuron activity form distinct clusters in principal component space. **j** Spike occurrence attributable to putative single neurons is stable across a 30-min recording session as visualized using example channel-projected principal components.

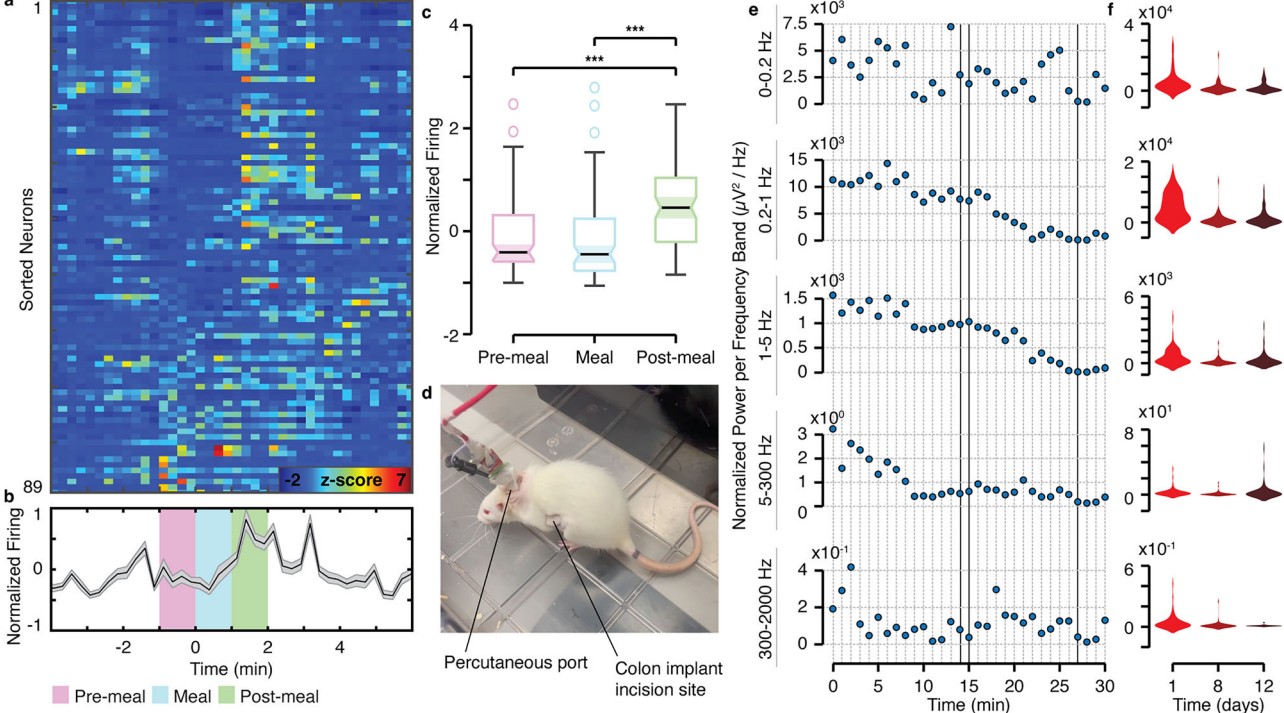

**Fig. 6 | Colonic activity is modulated by food ingestion and stress. a** Normalized instantaneous firing rate of all putative single neurons aligned to the onset of feeding and sorted by strength of pre-meal firing rate (warmer colors indicated higher z-scored firing rate modulation; *n* = 89 neurons, from 4 rats across 3 days of recording). **b** Average normalized instantaneous firing rate of putative single neurons during pre-meal (magenta), meal (cyan), and post-meal (green) epochs (shaded error bars represent standard error of mean; *n* = 89 neurons, from 4 rats across 3 days of recording). **c** Comparison of putative single neuron average normalized firing rate during pre-meal, meal, and post-meal epochs (Kruskal-Wallis ANOVA, $p = 4.25 \times 10^{-7}$; Wilcoxon signed-rank, pre-meal vs post-meal $p = 4.58 \times 10^{-6}$, meal vs post-meal $p = 1.90 \times 10^{-8}$; 89 neurons, from 4 rats across 3 days of recording; box center = medians; box edges = upper and lower quartiles; whiskers = non-outlier minimum and maximum; notch = 5% significance level). **d** Image of rat in

novel environment on Day 1. **e** Traces showing the temporal evolution of the normalized power per frequency band (μV²/H) per minute for the full duration of the chronic recording for rat 1 on day 1. The frequency bands used were: "0 - 0.2 Hz" to capture slow wave activity associated with interstitial cells of Cajal (ICCs)[60], originated locally in the submucosal plexus or potential volume-conducted components from deeper layers such as the myenteric plexus; "0.2–1 Hz" for slow rhythms such as those in circular smooth muscle[59,70]; "1–5 Hz" for faster rhythmic activities in smooth muscle[59,70]; "5–300 Hz" for primary skeletal muscle activity related to tissue and body movements (EMG related)[61]; and "300–2000 Hz" for high-frequency neural components[24,63–65]. **f** Violin plots showing the distribution of the normalized power per frequency range (μV²/Hz) within the first 15 min (data points) for all rats for each frequency band and day to visualize the decrease in the overall response along days.

microelectrodes, the effective recording volume for neural signals is within <50 μm radius[50]. As such, we positioned a series of 7 sets of 4 microelectrodes each in a linear fashion with an inter-set distance of 250 μm (Fig. 1a). This area combines for an effective measurement area of ~300 × 2000 μm, when considering the effective recording volume. We also included in the design a series of surgical markers, namely suture loops, to aid in chronic device placement, which are

visible as gold rings, identifying the pass-through for a suture (Fig. 1b). In addition to these features, we also included a series of further design considerations based on potential utility for this device moving forward in future studies. For example, we wired in 4 large 200 μm diameter electrodes, which are suitable for applied electrical stimulation[51]. For this study, we used these as surgical markers to aid in implant positioning, but these electrodes are

suitable for this application. We also arranged our microelectrodes into a tetrode configuration, which enables the identification of single units[50].

## Implant microfabrication & integration

Implants were constructed as previously published[23] and described as follows. Parylene-C was deposited using a dimerized precursor, dichloro-p-cyclophane (Specialty Coating Systems, Indianapolis, IN, USA), onto a Si wafer at 2 µm thickness. Subsequently, wiring was patterned using AZ 5214 photoresist (Microchemicals GmbH, Ulm, Germany) and developed using AZ 351B (Microchemicals GmbH, Ulm, Germany). Gold deposition was performed at 100 nm with an electron beam Evaporator (PVD 75, Kurt J Lesker) using 10 nm of Ti as an adhesion layer to the parylene-C substrate. Gold tracks were patterned using a lift-off technique in acetone. A second parylene-C layer was deposited at 2 µm to insulate gold tracks. Following insulation, a second photolithographic step was performed to pattern the outline of the device using AZ 10XT photoresist (Microchemicals GmbH, Ulm, Germany) and AZ 400 K developer (Microchemicals GmbH, Ulm, Germany). Using the photoresist as an etch mask, the outline of the device was etched down to the Si wafer using a reactive ion etcher (RIE) (PlasmaPro RIE 80, Oxford Instruments, UK). A third layer of parylene-C was deposited on top of the device with an intermediate layer of soap, used as an anti-adhesive layer to facilitate coating of electrodes with PEDOT:PSS. After parylene-C deposition, a final lithographic step was performed to pattern the openings for the electrodes and connector array using AZ 10XT and AZ 400 K developer, and the openings were etched using an RIE. Finally, a PEDOT:PSS mixture was spincoated onto the electrodes containing ethylene glycol (Sigma Aldrich, St. Louis, MO, USA), dodecyl benzene sulfonic acid (DBSA) (Sigma Aldrich, St. Louis, MO, USA), 3-glycidyloxypropyl trimethoxysilane (GOPS) (Sigma Aldrich, St. Louis, MO, USA), and PEDOT:PSS (Clevios PH 1000, Heraeus, Hanau, Germany). Patterning of the electrodes was accomplished using a peel-off technique. Finally, devices were removed from the Si wafer for bonding and integration with external electronic components. Implants were bonded (FINEPLACER pico2, Finetech GmbH, Germany) using an anisotropic conductive adhesive film (ACF) (10 µm particle size, 3T-TGP20500N, 3 T Frontiers, Singapore) to flexible flat cables (FFCs) using either commercial FFCs (Molex, Lisle, IL, USA) or a custom FFC (Fig. 5a) that we designed and produced for chronic measurements.

## Sample size calculations

As the data collected in this study is novel in nature and used to test the efficacy of the device produced in this study, specific power calculations for sample size were not possible. Instead, we focused on well-established sample sizes from other similar published studies focused on the development of devices. A minimum sample size of $n = 4$ rats was chosen for any statistically presented data to examine for reproducibility of results. In the case of mouse and pig data, no replications were performed. These data were utilized to test the surgical placement of the device and were not analyzed statistically.

## Acute rat surgery & electrophysiological measurements

All experimental procedures were performed in accordance with the UK Animals (Scientific Procedures) Act 1986 and were approved by the animal welfare ethical review body at the University of Cambridge. These procedures were performed under a project license (PP5478947) by surgeons working with personal licenses, A.J. Boys (I00597808) and A. Güemes (II0076024), issued by the UK Home Office. Female Sprague-Dawley rats (purchased at 200–250 g) (Charles River Laboratories, Kent, UK) were primarily used, except in studies assessing the scalability of the devices in mice and pigs, as described below. Rats and mice were group-housed in individually ventilated cages with ad libitum access to food and water for the duration of the

study and a 12 h light cycle (7 am–7 pm). Temperature in the facility was maintained between 20 and 24 °C with humidity between 45 and 65% in accordance with the UK Animals (Scientific Procedures) Act 1986. Animal sex was not considered in the design of this study, given that device performance was the main purpose of the study. The specifics of the surgical procedure varied with experiment, as is described above, but the following general protocol was used: Animals were anesthetized for surgery using isoflurane inhalant. A small incision was made into the peritoneal cavity to access the colon, typically along the flank of the animal, and an implant was positioned within the wall of the colon. A secondary incision was performed to place a ground into the subcutaneous space above the musculature on the contralateral side from the implant placement. The ground was positioned as distal as possible to avoid signal contamination from the heart rate. The ground consisted of a large area (~1 cm²) PEDOT:PSS-coated gold pad with poly(imide) backing. We note that ground placement plays a substantial role in noise reduction for these recordings and should be carefully positioned to improve recording quality. With the grounding setup and the general implant placement, we also performed a thorough experimental assessment of potential sources of noise and interference to ensure that we did not misclassify mechanical artifacts as an electrophysiological signal, specifically regarding the distension experiments, but also as validation for the pharmacological experiments. These validations involved repositioning of the ground, tissue palpation, electrode repositioning, etc. Recordings were performed using an Intan Headstage (Intan Technologies, Los Angeles, CA, USA) and an Intan RHS Stim / Recording System (Intan Technologies, Los Angeles, CA, USA). These data were collected at 30 kHz and were stored as Intan.rhs files, saved as 1-min segments. The first step in processing the data in Python (v3.10) was to combine all these segments into a single file, which was then stored as a Parquet file for efficient access and analysis. Injections were performed using a syringe pump (Legato®, KD Scientific, Holliston, MA, USA), and fluids were injected at a rate of 160 µL/s.

## Mouse study

All experimental procedures were performed in accordance with the UK Animals (Scientific Procedures) Act 1986 and were approved by the animal welfare ethical review body at the University of Cambridge. These procedures were performed under a project license (PP5478947) by surgeons working with personal licenses, A.J. Boys (I00597808) and T. Naegele (I68878875), issued by the UK Home Office. A female C57BL/6 J mouse (Charles River Laboratories, Kent, UK) was used for this study. Implanted devices were of identical construction to those used for rats. To access the colon, a laparotomy was performed, and the device was inserted into the colonic wall as it was in the rat study portion of this work. A subcutaneous ground was placed in a secondary location as above. The colon was distended as it was in the rat study portion of this work. Recordings were performed using an Intan Headstage (Intan Technologies, Los Angeles, CA, USA) and an Intan RHS Stim/Recording System (Intan Technologies, Los Angeles, CA, USA). Data analysis was performed in the same manner as was performed for the rat mechanical stimulation portion of this work, using a 4th-order Butterworth in order to visualize the low-frequency contributions of the smooth muscle response (0–300 Hz, lowpass filter with 300 Hz cutoff frequency), and the high-frequency activation contributions (300–2000 Hz, bandpass filter).

## Pig study

All experimental procedures were performed in accordance with the UK Animals (Scientific Procedures) Act 1986 and were approved by the animal welfare ethical review body at the University of Cambridge. These procedures were performed under project license (PP6076787) by surgeons working with personal licenses, S. El Hadwe (I54111393) and A. Carnicer Lombarte (IDA1CEDB9), issued by the UK Home Office.

A female pig (50 kg) (Large White / Landrace cross) purchased from (B&S Farming Ltd., The Heath, Woolpit, Bury St Edmunds, Suffolk, IP30 9RN) was used. The animal was positioned in a supine position on the surgical table. A midline incision was made in the abdomen, beginning just below the sternum and extending toward the pelvic region. The abdominal wall layers were carefully dissected to access the abdominal cavity, and the abdominal organs were gently retracted to expose the large intestine. Identification of the ileocecal valve and cecum facilitated the localization of the large intestine. A segment of the colon was selected and isolated from the surrounding tissues using sterile techniques, preparing the region for implantation. For the implantation process, reverse micro-forceps were introduced through the abdominal wall, creating a tunnel for the neural interface device. Once externalized, the distal end of the tunnel allowed precise insertion and guided "pulling" of the implant through the passage. A metallic wire was secured to nearby muscle tissue to ensure proper grounding for neural recordings. Implanted devices were of identical construction to those used for rats. For capsaicin administration, 500 nM capsaicin was poured over the colon for ~1 min while recording. Recordings were performed using an Intan Headstage (Intan Technologies, Los Angeles, CA, USA) and an Intan RHS Stim/Recording System (Intan Technologies, Los Angeles, CA, USA). For analyzing the impact on the tissue responses to capsaicin, the filtering process involved wide-bandpass filtering (0–2000 Hz) to visualize the complete frequency range of interest. The spectrogram of a window of 480 s after the addition of the drug to the tissue was plotted to analyze the temporal and frequency responses of the tissue to capsaicin.

## Ex vivo preparations & electrophysiology

Experiments using animal tissue were conducted on animals euthanized in accordance with Schedule 1 of the UK Animals (Scientific Procedures) Act 1986 Amendment Regulations 2012 under the following approval from the University of Cambridge Animal Welfare Ethical Review Body. Adult male C57Bl/6 mice (10-12 weeks) were obtained from Charles River (Cambridge, UK; RRID:IMSR_JAX:000664). Mice were housed in temperature-controlled rooms (21 °C) with a 12-h light/dark cycle and provided with nesting material, enrichment (e.g., tubes, chewing blocks and shelters) and access to food and water *ad libitum*. Animals were euthanized by rising concentration of $CO_2$ or cervical dislocation, followed by exsanguination. Electrophysiological recordings of lumbar splanchnic nerve (LSN) activity were conducted as previously published and described as follows[52,53]. Briefly, the distal colorectum (splenic flexure to rectum) and associated LSN (rostral to inferior mesenteric ganglia) were isolated from euthanized mice as described above. The colon was cannulated with fine thread (Gutermann) in a rectangular Sylgard-coated recording chamber (Dow Corning, UK) and bath superfused (7 ml/min; 32–34 °C) and luminally perfused (100 μL/min) by a syringe pump (Harvard apparatus, MA) against a 2–3 mmHg pressure with carbonated Krebs buffer solution (95% $O_2$, 5% $CO_2$) to keep the tissue alive. Krebs buffer was supplemented with 10 μM atropine and 10 μM nifedipine to prevent smooth muscle activity. Borosilicate suction electrodes were used to record the multi-unit activity of LSN bundles. Signals were acquired at 15 kHz at 20 kHz (Micro1401; Cambridge Electronic Design, UK) and exported to txt format with Spike2 (Cambridge Electronic Design, UK). Preparations underwent a minimum 30-min stabilization period to ensure baseline firing was consistent.

## Drug preparation and administration

Capsaicin (Tocris Bioscience, Bristol, UK) was dissolved in ethanol to yield a 1 mM stock solution. The required volume of this stock solution was then diluted in 20 ml of Krebs solution, resulting in a test solution with a concentration of 500 nM. Bradykinin (Tocris Bioscience, Bristol, UK) was dissolved in distilled water to obtain a 10 μM stock solution. The necessary volume of this stock solution was subsequently diluted

in 20 mL of Krebs solution, yielding a final test solution with an adjusted concentration of 1 μM. Topical administration consisted of pouring drops at the location of the implant on top of the colonic tissue, whereas intraluminal doses consisted of delivering the solution into the adjacent luminal cavity using a syringe needle and a pump.

## General analytical methodology

The raw signals undergo a sequence of filtering and characterization processes in the frequency domain, tailored to optimize the analysis for individual objectives and the collected data set. In general terms, each raw signal was first filtered using Butterworth filters of order 4 for high-frequency or low-frequency noise reduction (lowpass and band-pass filters, respectively), with specific cutoff frequencies detailed in the relevant subsection for each experimental condition. Notch filters at 50 Hz and odd harmonics were applied when electrical contamination was significant and specified in the respective section. For all the signals, the power spectral density (PSD) of the signals was computed using the Welch method with a segment length of 512 samples (non-overlapping). The Welch's method computes an estimate of the PSD by dividing the data into segments, computing a modified periodogram for each segment, and averaging these periodograms. This approach ensured a detailed analysis of frequency components while retaining the integrity of the original signals. Color map representations known as spectrograms were generated using the Python function plt.specgram. These visualizations were meticulously set to 'psd' mode, showcasing frequencies with a decibel ('dB') scale, ensuring an exact portrayal of frequency details. Spectrograms, unlike a simple Power Spectral Density (PSD) chart, offer a dynamic visual representation of how frequencies change over time in the signals. While PSD shows the strength of different frequencies at a single moment, spectrograms go further, illustrating how these frequencies evolve and interact across time. The methodology further involved the meticulous selection of specific windows of interest within the signal, enabling targeted and comprehensive investigation of crucial segments relative to each stimulation trial. The PSD and spectrograms for each of these segments were also computed, providing detailed frequency analyses to enhance the understanding of the data within these selected windows. The subsequent sections delve into the precise details of analysis and the criteria used to select parameters tailored to address each distinct experimental condition.

## Analysis of drug stimulation trials

To analyze the impact on the tissue responses to different chemical stimuli (i.e., drugs), the filtering process involved two strategies: wide-bandpass filtering (0–2000 Hz) to visualize the complete frequency range of interest, and tight-bandpass filtering (300–2000 Hz) with the addition of a notch filter to remove electrical noise and isolate the high-frequency components of the signals. For the rest of the analysis, only the tight-bandpass filtered traces were used. Windows of 10 s before and 90 s after the addition of each drug to the tissues (locally or intraluminally) were extracted to analyze the temporal and frequency responses of the tissue to each drug. The analysis was divided into two strategies. To evaluate temporal changes in signal frequency characteristics in each extracted window, a rolling AUC computation method was employed using spectrogram data. The analysis involved iterating through columns of the power spectrum using a non-overlapping rolling window approach. The window size was calculated as the number of samples within the 1 s in the window. The AUC within each window of the spectrogram was carried out through numerical integration using the scipy.integrate.simps function from the SciPy library, and was plotted over time. Key metrics of the resulting temporal AUC, such as the maximum, the mean and the variance of the AUC, were computed and reported. These metrics offered insights into the dynamic changes and variability of signal characteristics within distinct temporal segments derived from the spectrogram analysis.

For easy comparison across all injection experiments, each metric was normalized across the various drug types, guaranteeing their values fell within the 0-1 range for each experiment. Barplots were employed to illustrate the response based on the metrics to the four distinct drugs (BK, serosal capsaicin, intraluminal capsaicin, and wash-control). Additionally, in the recording featuring multiple BK additions, both consecutive and separated by wash additions, barplots of the metrics were computed to evaluate desensitization. Secondly, the Power Spectral Density (PSD) was derived from the computed power spectrum data obtained from the spectrogram at distinct time points: before the drug was introduced, immediately after its administration, and at the conclusion of the response. This approach was employed to offer a more comprehensive visualization, allowing for a clearer understanding of how frequency components evolved over time in response to the drug.

### Analysis of contraction traces

A time window spanning 30 s before and 160 s after the visual initiation of the contraction was isolated across all channels. Similarly to the previous section, a wide and tight bandpass traces were extracted, and a notch filter was applied. The spectrogram of the wide-bandpass filtering (0–2000 Hz) was plotted. From the tight-bandpass filtered traces (300–2000 Hz), the envelope of these signals was calculated to visualize the signal amplitude's progression during the contraction more clearly. This involved using NumPy's Hilbert transform method to generate an analytic signal, from which the absolute value was derived to establish the signal's amplitude envelope. To enhance this envelope and minimize noise, a smoothing procedure was implemented using a 1 s rolling average window via convolution.

### Analysis of distension stimulation

The distension dataset underwent two 4th order Butterworth filters: 1) low-pass filter with cut-off frequency at 300 Hz, chosen based on empirical evidence of activation of smooth muscle in the gut to be lower than 200–300 Hz[54], and 2) tight-bandpass filtering (300–2000 Hz). A notch filter was not needed as the electrical noise was negligible. For the low-frequency bandpass signals, a window segment of 10 s after each distension was extracted for further analysis. The Area Under the Curve (AUC) was calculated for the positive part of all the extracted windows (low-pass filtered voltage traces) using the trapezoidal rule (*trapz* method from the numpy library) for numerical integration. To enable consistent comparison across experiments concerning the four distention responses and particularly the variations pre- and post-increase in isoflurane, the normalized AUC (nAUC) was computed. This metric reflects the AUC for each distension segment, standardized by subtracting the minimum value and dividing by the range of the AUC observed across segments within each experiment (0-1). This normalization approach ensures a fair assessment of the responses across experiments. Statistical analyses were conducted using Python libraries scipy.stats, primarily employing two unpaired two-sided key tests: the t-test and the Mann-Whitney U test. These tests were applied for unpair comparisons between different groups or conditions within the dataset. The t-test was utilized to assess the significance of differences between means when the data met the assumptions of normality and homogeneity of variance, checked respectively using Shapiro-Wilk (for normality) and Levene's test (for homogeneity of variance). Conversely, the Mann-Whitney U test was employed for non-parametric analysis when the assumptions for the t-test were not met. Effect size Cohen's d was calculated as the difference between the group means divided by the pooled standard deviation of the two groups.

### Chronic surgery & electrophysiology

Prior to surgery, rats were pre-conditioned to eat a commercial hazelnut-cocoa paste (Nutella®) for 2 weeks. Rats were group-housed in individually ventilated cages with ad libitum access to food and water for the duration of the study. Animal sex was not considered in the design of this study given that device performance was the main purpose of the study. For chronic surgeries, female Sprague Dawley rats were anesthetized using an isoflurane inhalant. Rats were given a pre-operative analgesic (Carprofen, Carprieve® Injection, Norbrook Laboratories) with a saline injection for fluid replacement. A primary incision to the subcutaneous musculature was made into the right flank of the rat approximately 1 cm distal to the rib cage. A secondary incision between the shoulder blades was made to the subcutaneous musculature. A tunnel was formed subcutaneously between these two incisions using blunt dissection, and the implant wiring was threaded through the tunnel, leaving the microfabricated portion of the implant on top of the musculature in the primary incision. Using a modified transcutaneous port (Vascular Access Button for rat, Linton Instrumentation, UK), stainless steel wire was threaded into the anchoring sponge of the port to form the ground for the subsequent recordings. The port was fitted into the secondary incision, the implant wiring was run through the port, and the pass through the port was back-filled with medical grade silicone (Kwik-Cast silicone, World Precision Instruments). The secondary incision was then sutured closed, and a Protective aluminum magnetic cap (Linton Instrumentation, UK) was placed over the port to prevent chewing on exposed wiring. At the primary incision, subsequent incisions were made through the subcutaneous musculature to access the peritoneal cavity. The colon was located in the vicinity of the incision, and the implant was placed into the wall of the colon, using the technique described above (Fig. 1d through h). The end of the implant was sutured into the wall of the colon using 9–0 non-resorbable sutures. The incision site was sutured in a layered fashion to bind the different layers of musculature around the implant wiring, using the interconnects that we designed into the implant wiring to anchor this peritoneal crossing in place. Finally, the primary was sutured closed, and the rats were allowed to recover. Rats were given post-operative analgesia each day (meloxicam, Metacam®, Boehringer Ingelheim Labs), through 3 days post-surgery. For the first subset of rats ($n = 2$), recordings lasting over 1 h were performed at Days 1, 7, and 14 to assess stability of the implants. At this point, the rats were sacrificed, and the second rat's colon was examined and removed for histological preparation (Supplemental Fig. 9). The second subset of rats ($n = 4$) was used for behavioral experiments. Rats underwent training prior to the surgery to consume a palatable food, Nutella® (3 g). To standardize recording conditions, food was removed at 8 am on the day of recording, allowing for a minimum 4-h period of food restriction. Each recording session over a period of up to 14 days was initiated by connecting the device to the acquisition system and moving the rat to a new environment (new cage) leading to stress response (Supplementary Movie 4). After 15 min, Nutella® was placed into environment recorded continued for at least 15 min after feeding. Electrophysiological recordings were obtained on Days 1, 8, and 12 following device implantation.

### Single unit isolation & spatiotemporal analysis

Recordings were visually inspected for quality using Neuroscope. Epochs featuring synchronized high-amplitude sustained high-frequency activity consistent with EMG contamination[55], along with sharp voltage deflections indicative of mechanical/electrical noise, were manually identified and excluded from downstream MATLAB analyses. Recordings were high-pass filtered at 300 Hz. For each recording session, the 5 longest artifact-free epochs were used for each channel to calculate the noise floor, $noise\ floor = mean(\frac{|data|}{0.6745})$[24]. A multiple of this noise floor (4–6, as curated by individual session data quality) was used as the threshold for detection of extracellular spikes. Negative peaks with amplitude greater than the threshold were detected as action potentials. Coincidental spikes across all channels were deemed artifacts and removed from downstream analyses.

Detected spikes were visually inspected alongside filtered raw recordings for veracity. Isolation of putative single neurons was done by inputting high-pass filtered recordings and detected spikes into KiloSort, and the resulting clusters were visualized in Phy2. Clusters exhibiting physiologic refractory period, waveform characteristics and spatial profile were included in downstream analyses[56]. Auto-correlograms and cross-correlograms were obtained with temporal convolution of single-unit spike trains[57]. Repeated temporal shuffling of time bins were used to obtain 95% confidence intervals. Principal component weights calculated by KiloSort were used to demonstrate clustering in principal component space and stability over the course of a recording session[58].

A signal was classified as originating from a "putative single neuron" only if it met several strict criteria simultaneously: 1) distinct and characteristic waveform shape typical from neural signals (e.g., asymmetrical); 2) The neuron's firing pattern exhibited a refractory period, meaning it was impossible for it to fire another action potential for a brief period after the first. This is a hallmark of neuronal activity; 3) The spike was largest on one electrode and decayed rapidly on adjacent electrodes, consistent with a localized, point-source origin (i.e., a single cell body); 4) In principal component analysis, the waveforms from a single putative neuron formed a distinct and stable cluster, separable from noise and other units.

### Feeding electrophysiology analysis

Putative single neuron spiking was z-scored across each recording session to obtain normalized spiking rate patterns. Individual neuron spiking rates were calculated in 3 s bins, aligned to the onset of feeding, and visualized across the pre-meal, meal, and post-meal epochs. Use of bins ranging from 1–15 s did not alter conclusions. Population neural spiking activity was obtained by averaging the responses of all individual neurons in each epoch. Kruskal-Wallis ANOVA with post-hoc Wilcoxon signed-rank test and Bonferroni correction was used to determine statistical differences in population neural spiking between the epochs.

### Stress-response multifrequency band analysis

The purpose of this multifrequency band analysis was to gain insights into the contributions of various biological sources to the recorded signals in the context of stress response. To start, we computed the average and standard deviation of the impedances for all working electrodes on each recording day and compared these metrics across different days. Channels that remained active across all recorded days were selected for further analysis. Signals from these selected channels were initially downsampled to 10 kHz to facilitate efficient processing. To minimize shared artifacts, signals were referenced to a common baseline; specifically, the mean signal across all selected channels was computed and subtracted from each channel's signal, producing a set of referenced signals. These referenced signals were then segmented into 1-min intervals to analyze the temporal evolution of the signals. Within each interval, we divided the data into predefined frequency bands and applied bandpass filters to isolate these bands for detailed examination. The frequency bands used were: "0–0.2 Hz" to capture very low-frequency components and slow wave activity associated with interstitial cells of Cajal (ICCs), originated locally in the submucosal plexus or potential volume-conducted components from deeper layers such as the myenteric plexus; "0.2–1 Hz" for slow rhythms such as those in circular smooth muscle; "1–5 Hz" for faster rhythmic activities in smooth muscle; "5–300 Hz" for primary skeletal muscle activity related to body movements; and "300–2000 Hz" for high-frequency neural components. The frequency bands were chosen based on findings from electrogastrography (EGG)[59], research on interstitial cells of Cajal (ICCs) in ex vivo settings[60], and studies of electromyography[61,62] and extraneural activity[24,63–65]. Power Spectral Density (PSD) for each channel was computed using Welch's method, with a segment length of 1,000,000 samples to achieve a resolution of approximately 0.01 Hz, necessary for resolving the very slow frequencies of ICC cells. An overlap of 50% between segments was applied. The power within each frequency band was normalized by the bandwidth to allow for fair comparison across bands of different widths, labeled as normalized power per frequency range ($\mu V^2/Hz$) in the text. Visualizations were created to compare the evolution of power in each frequency band over time for each animal. Boxplots summarized the power profiles for all rats. Violin plots were also used to visualize the distribution of the normalized power per frequency range within first 15 min for all rats for each frequency band and day to visualize the response trend along days.

### Ethics

Each experiment involving animals, human participants, or clinical samples have been carried out following a protocol approved by an ethical commission.

## Data availability

All data supporting the findings of this study are available within the article and its supplementary files. Any additional requests for information can be directed to, and will be fulfilled by, the corresponding authors. The main data generated in this study are publicly available under a Creative Commons Attribution 4.0 International license in the Zenodo database under accession codes https://zenodo.org/records/15162113[66] (Part I, V3) and https://zenodo.org/records/15162121[67] (Part II, V1). Source data are provided with this paper.

## Code availability

The analysis code (Version 1), and instructions to install and use have been made publicly available in a version-controlled repository on GitHub and archived via Zenodo (https://doi.org/10.5281/zenodo.17142906) under the AGPL-3.0 license[68]. All scripts developed in Matlab and Python used standard scientific libraries (numpy, pandas, scipy, matplotlib, and seaborn) and can be used to reproduce the analyses reported in this study.

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

## Acknowledgments

The authors would like to acknowledge Sam Hilton for supporting animal work, and Sagnik Middya for his advice in backend electrical design and implementation. The authors would also like to acknowledge MRC Metabolic Diseases Unit (MC_UU_00014/5) for histological processing of paraffin-embedded sections. AJB acknowledges support from the HFSP Cross-disciplinary Fellowship [LT000034-C/2020]. AGG acknowledges support from the Royal Commission for the Exhibition of 1851 Research Fellowship, the Royal Academy for the Engineering Research Fellowship [RF-2324-23-284], and Rosetrees Research Fellowship. RAG acknowledges support from the Cambridge Trust Fellowship. TN acknowledges support from the EPSRC IRC in Targeted Delivery for Hard-to-Treat Cancers [EP/S009000/1]. FU acknowledges support from the Marie Skłodowska-Curie Action COFUND Program (APEX) [754535] and the Science Foundation Ireland [22/PATH/S-10882]. NPH acknowledges support from the Science Foundation Ireland [12/RC/2273_P2]. GMM acknowledges support from the NIHR Cambridge Biomedical Research Center [NIHR203312]. RMO acknowledges support from the US Air Force Office of Scientific Research (AFOSR) [FA8655-20-1-7021]. The views expressed are those of the author(s) and not necessarily those of the NIHR or the Department of Health and Social Care. For the purpose of open access, the authors have applied a Creative Commons Attribution (CC BY) license to any Author Accepted Manuscript version arising from this submission.

## Author contributions

Conceptualization: A.J.B., A.G.G. and RMO Methodology: A.J.B., A.G.G., R.A.G. and F.U. Investigation: A.J.B., A.G.G., R.A.G., Z.L., C.L., S.E., A.C. and T.N. Formal analysis: A.J.B., A.G.G., L.M., J.N.G. and D.K. Visualization: A.J.B., A.G.G. and L.M. Code: A.G.G. and L.M. Funding acquisition: A.J.B., A.G.G., D.C.B., G.G.M. and R.M.O. Project administration: R.M.O. Supervision: D.G.B., D.C.B., N.P.H., G.G.M. and R.M.O. Writing – original draft: A.J.B. and A.G.G. Writing – review & editing: A.J.B., A.G.G., L.M., R.A.G., Z.L., C.L., S.E., A.C., T.N., F.U., D.G.B., D.C.B., J.N.G., N.P.H., D.K., G.G.M. and R.M.O.

## Competing interests

The authors declare no competing interests.
