## [Transparent Peer Review file · Nature Communications]

Implantable Bioelectronics for Gut Electrophysiology

Corresponding Author: Professor Roisin Owens

Version 0:

Reviewer comments:

Reviewer #1

(Remarks to the Author)

This is an excellent study of using gold based electrodes to measure electrophysiological signals arising from enteric nervous plexi. I am very impressed with the surgical considerations involved in navigating the complex twists and turns of the intestines without harming the intestines. I have some questions.

- 1) I am not familiar with the ENS and the enteric nervous plexi in humans. But how long must you construct your entire monitoring device in order to scale this up to the human? How difficult would it be to ensure no breaks in the wiring along such a long device?
- 2) where would be the exit point of the device for the human? Would that be at the back like in the mice or in front? These may have implications if the patients monitoring period is over at least one cycle of wake and sleep since most humans lie on their backs while sleeping.
- 3) how long do you expect the monitoring to be for humans? in your experiments, I believe you have monitored for a maximum of 3 days. Would your device be able to survive and be still functional within the intended time of monitoring?
- 4) what is the current gold standard of monitoring electrophysiological signals of the ENS? is it possible to just pick one of three situations and also monitor using the gold standard method and compare the findings? all of the experiments are assumed that the leads are measuring what it was intended to measure without a comparative control to ascertain whether the signals picked up by your device is the true signal. If the gold standard method is very difficult, perhaps, you can use your device to measure more easily accessible plexus where both your device and the gold standard method can be used and then compared?

(Remarks on code availability)

I do not have any understanding of coding. Sorry.

Reviewer #2

(Remarks to the Author)

The authors showed the construction and validation of a bioelectronic device to access neural information produced and processed in the gastrointestinal tract. It is interesting. However, one of the major questions is that why you used the flexible electrode rather than stretchable electrode, as we can see the clearly slide between the electrode and the gut in the attached video.

1. In fig.4, you should also give the scale bar of the spike amplitude. In fig.4f, why you only have 20 recordings rather than 28 (in Fig.1, you have 28 recording electrode).
2. The pig experiment is also important, why you don't show it as the rat in Fig.4?
3. For the Implant Microfabrication & Integration, you don't mention any encapsulation, or you just forget it?
4. You should give the device characterization of the electrode, ie EIS.
5. I think the major achievement of this study is the recording of putative single neurons. As the authors mentioned, it is putative and you need to prove it. It is confusing if you want to publish it on Nature Communications when you don't have enough data. And you don't show the pig and mouse data.

(Remarks on code availability)

Reviewer #3

(Remarks to the Author)

This is an exciting study from Boys et al. developing a novel technique to record electrophysiological activity in the gut in vivo in free-moving rats. The authors develop a flexible, implantable device and a surgical procedure that enables recording from the enteric nervous system (ENS) while allowing distension and contraction movements of the gut wall. To examine neuronal activity, the authors first performed recordings in vivo under anaesthesia, apply bradykinin and capsaicin to the gut wall during recording, and investigated high vs low frequency responses. In addition, the concentration of isoflurane anaesthesia was investigated. Further, the authors examined activity in free-moving rats, recording activity from what they identify as individual neurons, before, during and after feeding. They also examined the impact of stress on neuronal activity. In addition to rats, the authors have also adapted these implants for larger (pigs) and smaller (mice) animal models.

This is a very exciting and novel advance in this field, enabling researchers to record from the ENS (as well as other electrophysiologically active cells in the gut) in live, free-moving animals. However, there are several major concerns regarding the results.

1. The biggest concern is confirmation of whether the signals that are recorded are arising from enteric neurons. As the authors state, there are multiple types of electrically active cells in the gut wall, including enteric neurons, interstitial cells of Cajal (although they play a smaller role in the colon compared to the stomach), as well as the smooth muscle cells. The device generated by the authors essentially performs extracellular recordings, and while the authors state that they have produced single-unit recordings from individual neurons, the data that is shown does not definitively prove this. Can the authors provide more evidence that the recordings that they identify as "neuronal" do indeed arise from neurons? A vital experiment would be to perform recordings using this device coupled with simultaneous recording from enteric neurons (either intracellular recording or calcium imaging), in an ex vivo setting. This has previously been achieved using a somewhat reliable recording device (Athavale et al., 2025 PMID 38860855). The authors also use high and low bandpass filters to specifically show high and low frequency activity. It is unclear why they did this, and how this relates to the identification of neuronal activity. Acknowledging my lack of familiarity on the engineering aspect of this, some explanation and/or references on this filtering and signal identification would be helpful to the neurogastroenterology audience. The lack of clarity on this vital point shadows the various in vivo experiments that were performed, such as stress and feeding-associated changes in activity. If we cannot be sure that these responses do indeed come from enteric neurons, it becomes difficult to extrapolate and interpret the data from these experiments.

2. The authors state that they are recording from the submucous plexus of the ENS. They state that the device is placed "underneath" the muscularis external (page 2), and the recording electrodes directed lumenally. However, from the figures shown, it's difficult to confirm this. From Fig 1 and Supp fig 1, the location of the "tunnel used for implant placement" is shown. However, depending on the orientation of the tissue when it was sectioned (transverse or longitudinal), it is extremely difficult to identify which cell layers are around this tunnel. Presumably, the device was removed before tissue processing and histological analysis, hence its precise location is difficult to ascertain. As a first point, some higher magnification H&E images would be helpful. In addition, some immunohistochemical (or immunofluorescent) staining of the tissue to distinguish between the different cell types and also identify the location of enteric neurons is needed. A further point is that annotation of the images to help orient the reader would be beneficial for many of the figures (please see specific comments below).

Lastly, throughout the manuscript, the authors refer to recording electrical activity that is more related to the function of myenteric neurons, and myenteric and muscular ICC activity. Therefore it is key to identify whether the device is indeed recording from submucous plexus activity, or the external muscle (plus myenteric) layers.

3. To strengthen the study and its implications, the authors may find it beneficial to collaborate with researchers or experts who have extensive experience and specialised knowledge in the enteric nervous system and gastrointestinal physiology. There are several instances throughout the manuscript where more specificity with wording and references, as well as more detailed information on the anatomy and neurophysiology is needed (please see below for specific comments).

4. More thorough proof-reading of the manuscript would be helpful. There are many small errors in this manuscript, although individually these are all very minor, the cumulative effect is very distracting. For example, it would have been helpful to see greater consistency around the use of acronyms, and also greater consistency around formatting - some words are italicised while others are not. The use of line and page numbers would have also helped with the review process.

Specific comments by Section:

Summary:

5. Sentence 4: The authors use the phrase "the constant intrinsic and extrinsic motion" of the gut. What do they mean by extrinsic motion? There is extrinsic innervation of the gut, but generally, there is not much reference to "extrinsic motion" of the gut.

6. Final sentence: ENS has already been used as an acronym already used prior to this point

Introduction:

7. Paragraph 2, sentence 1: the authors state that the ENS plexi "wrap circumferentially around the GI tract", this phrasing

potentially discounts the continuity of the plexi in the longitudinal direction. The ENS in healthy conditions spans the entire length of the gut as well as spanning the circumference.

8. Paragraph 2: the authors state that past recordings of the ENS have focused on ex vivo recordings with “patched electrophysiological setups”, presumably they mean patch-clamp recordings from ex vivo preparations of the gut. However, this is not strictly accurate as there are very few studies that have used whole-cell patch-clamp to investigate the activity of enteric neurons. Most studies have used sharp electrode intracellular recording. While this may seem like a small point of difference, it highlights a lack of familiarity with the literature in this field.

9. Paragraph 2, last sentence: The authors state that there are “no current devices demonstrating successful in vivo recording of ENS electrophysiology”. However, there are some studies that have achieved this, for example <https://doi.org/10.1152/ajpgi.00069.2024> uses flexible multielectrode arrays to record electrical signals in the rodent stomach in-vivo. There is also evidence of recordings from the porcine stomach PMID: 28035728 and 35726361, and in human stomach <https://doi.org/10.1152/ajpgi.00125.2010>. The animal studies have been performed under anaesthesia, therefore the current work is highly novel and a great advance to be able to perform these recordings in conscious, free-moving animals. However, recognition of the previous studies should also be taken into account.

Implant Design & Surgery Development

10. Figure 1d- 1i: These panels would benefit with some more labelling to orientate the reader. Panel d-i, labelling of the small intestine vs colon would be helpful, as well as other structures that are visible. Also why are the surgical images in this figure so pink? It seems the contrast has been altered significantly

11. Figure 1 legend: There is reference to a red box when discussing the loops designed for suturing the device however there is a red box around panel j which has no relevance to this. Please check what you are trying to refer to in the figure and update the legend accordingly.

12. Figure 1j: additional landmarks need to be added to this panel, it is incredibly difficult to orientate the key features. It does seem like the cavity where the electrode was is between some muscle layer and the muscularis mucosa which would indicate the submucosal space however if the recording interface was facing internal there is no evidence in this image of any neurons being present in that space. The authors might look to a better representative image being used here. What anatomical plane were the sections taken? Transverse across the full circumference of the tube?

Distension of gut wall validates physiological responses to mechanical stimuli:

13. Paragraph 1, first sentence: the authors describe a lot of physiology in this first sentence, but cite a reference that mostly focuses on the neurochemical subtypes of enteric neurons. There are many other more physiologically relevant studies and reviews that could be used here, including PMID 32152479 (which is already cited).

14. Supplementary video 3: The tissue looks very dry in this video. Were the authors able to maintain tissue integrity through these experiments?

15. Paragraph 2: the authors state that in response to mechanical distension of the gut, they see “asynchronicity of the response” which implies that multiple cell types are responsible for the mechanical distension. And further suggest that ICCs which are “crucial intermediates in the GI tract for bridging neurons to muscle tissue” have contributed to the recorded activity following distension. There are many issues with this.

Firstly, colonic motility is mostly neutrally mediated, not myogenically mediated. Indeed, ICC are interconnected within this network but they are not essential while the neuronal network is. Therefore, the dominant signal here should be from the neurons. While there may be a slow wave component that involves ICCs, these are not as prominent in the colon as they are in the stomach. Therefore, the authors should presumably expect to see a dominant frequency of activity from neurons, instead of a series of asynchronous spikes? The citation of reference #25 has a strong focus on gastric motility networks, which is very different from that of the colon. For the colon the critical component of this network for motility is the ENS not the ICC.

Secondly, the authors previously state that they have oriented the recording device so that activity from the myenteric plexus and smooth muscle layers, and so presumably myenteric and muscle associated ICCs are not being recorded. However, this contradicts the current statements that signals from ICCs are contributing to the distension-evoked responses. Given the placement of the electrodes interfacing with the submucosal plexus where there are less neurons and less ICC, there is potential that single spikes are that would be detected. An additional complication to this is that submucosal neurons in rodents are generally less involved in control of gut motility. While they may respond to distension-evoked activity, it is uncertain what is the importance of detecting their signals with this mechanical stimuli.

16. Figure 2: Additional labels would help orient the readers

17. What are the benefits of the high and low bandpass?

Pharmacological stimuli initiate multi-frequency electrophysiological response with common neurophysiological characteristics

18. Why was bradykinin and capsaicin topically administered? In particular, this is not a relevant stimulus for the submucosal plexus. We recognise that technically this was a more straight-forward route of administration, however, intraluminal administration (which was provided in the case of capsaicin) would have been more physiologically relevant.

19. Could the authors also provide some more information on the topical administration route, ie. was there any shielding of extrinsic neural pathways within the abdominal cavity that may influence these signals that are being identified?

20. Could post-hoc analysis of the plexus neurons in the implant region be completed to identify the chemotype/phenotype of the neurons being recorded from?

21. Do the authors have any immunohistochemical staining of the plexi/neurons post implantation to investigate damage/integrity?

Conformable devices enable to record colonic responses to feeding and stress

22. Why is it necessary to filter to capture slow waves from the submucosal plexus? There will be no slow waves generated

from ICC-submucosa this is a ICC-myenteric and ICC-intramuscular phenomenon. Please see point above. While it seems across this manuscript their aim is to record from the submucosal ganglia not the myenteric ganglia but there are references to activities that are controlled primarily from the myenteric ganglia when these neurons are not the ones being recorded from?

Conclusion

23. Final sentence: while we appreciate that this newly designed flexible array provides a great advance in terms of being able to record from free-moving, in vivo animals, recognition of previous work where recordings were performed in vivo is also important. Greater familiarity with these references can perhaps help the authors interpret their signals and also hone in on the question of which responses correlate directly with ENS activity.

Methods

Device Design

24. Sentence 2: the inter-ganglionic distance mentioned here and in the referenced paper is for myenteric plexus not the submucosal plexus. The submucosal plexus is sparser than the myenteric and would have larger inter-ganglionic spaces

25. Sentence 11-12: It is stated that the arrangement of the microelectrodes would "theoretically enable the identification of single units. However, we did not utilize this capability in this study." But the authors state in the paper that they obtained single neuron data. Are these different things?

26. Again, in the final sentence the ENS acronym has already been used

Implant Microfabrication & Integration

27. Sentence 3: grammatical error an

Drug preparation and administration

28. Sentence 3: grammatical error of instead of on

Analysis of drug stimulation trials

29. Re-word sentence 5 grammatical error

Chronic surgery & electrophysiology

30. Last sentence: need to reword this sentence

Supplementary

Figure 2

31. The figure legend says urethane, but the data shows only high and low isoflurane?

Figure 6

32. The data in the figure shows mouse, rat and pig but the title only includes mouse and pig

(Remarks on code availability)

n/a

Reviewer #4

(Remarks to the Author)

(Remarks on code availability)

Version 1:

Reviewer comments:

Reviewer #1

(Remarks to the Author)

The authors have sufficiently addressed all of my concerns. It is an impressive work and I believe the field would benefit greatly from this work. Congratulations, authors.

(Remarks on code availability)

Sorry, I don't have enough expertise in assessing coding.

Reviewer #2

(Remarks to the Author)

1. It's not a good answer regarding the use of a flexible vs. stretchable device. If you can place the flexible device into the colonic wall, you can do the same for a stretchable device.
2. As the author mentioned that the device includes 28 recording channels, as correctly noted. Of these, 20 channels maintained impedance values within our predefined range for high-quality neural recordings. The remaining channels exhibited elevated impedance and were excluded from analysis. Please think about why? The reason is that there is a clear slide between the electrode and the gut. Then the 8/28 electrodes cannot contact with the colonic wall, which resulted in high impedance.
3. Regarding the general anesthesia leads to a "strong suppression of spontaneous neural activity," it is easy to reduce the anesthesia level then you can get a good signal..
4. For the Implant Microfabrication & Integration, you don't mention any encapsulation. What I mean is that how you encapsulate the flexible electrode with the cable or connector. What you talked about is Microfabrication, not Integration.
5. For the recording of putative single neurons, there are a lot of parameters that can be used to characterize the "spike". Please refer Moore et al., Science 355, eaaj1497 (2017).

(Remarks on code availability)

Reviewer #3

(Remarks to the Author)

We thank the authors for carefully addressing our concerns. We understand the difficulties with some of the proposed experiments such as simultaneous ex vivo gut recordings with this novel bioimplantable device, and these have been clearly described by the authors. We also appreciate the additional information on signal filtering and clarification on device placement and orientation. This novel device presents an exciting development in the ability to record activity from the ENS in conscious, free-moving animals.

One minor point:

The current device is recording from the submucous plexus, and we appreciate how the authors have carefully addressed that while most electrical signals from the myenteric plexus have been attenuated, some information from the myenteric neurons, ICCs and outer smooth muscle cells can still be detected, and also, information is likely to be relayed to via submucous neurons. Given the importance of the myenteric plexus in the control of gut motility, how easily can this device be adapted to record from the myenteric plexus instead of the submucous plexus? Could the authors make a comment on this?

(Remarks on code availability)

Reviewer #4

(Remarks to the Author)

(Remarks on code availability)

Response to Reviewers

We would like to thank the reviewers for their comments and thorough analysis of our manuscript. Overall, we have added and edited substantial text in the manuscript, added new experimental data as well as two new figures to the extended data and 12 new references. We feel that this process has greatly improved the overall quality of our presented work and enabled us to confirm and further explicate some points that were confusing or difficult to follow in the initial iterations of our work.

Reviewer #1:

This is an excellent study of using gold based electrodes to measure electrophysiological signals arising from enteric nervous plexi. I am very impressed with the surgical considerations involved in navigating the complex twists and turns of the intestines without harming the intestines. I have some questions.

1) I am not familiar with the ENS and the enteric nervous plexi in humans. But how long must you construct your entire monitoring device in order to scale this up to the human? How difficult would it be to ensure no breaks in the wiring along such a long device?

Thank you for this question. We performed our demonstration of the device in the pig model (see Fig. 4 - moved from Extended Data to the main text) to assess the feasibility of using this device in a model that is scaled to the human gut. The enteric nervous system scales in cell number with animal size, but the density maintains the same order of magnitude (Gabella, *Neurosci*, 1987). Additionally, the pig gut is considered to be quite similar to the human gut (Li et al., *Comm Bio*, 2023), but this is still an active area of study. The layout of our device is designed to regionally scan electrophysiology of the gut. While placing an implant down the length of the GI tract would be desirable, the size of such a device, even for the rat, would be surgically infeasible given the ENS extends from the esophagus to the rectum. If one were to design a study targeting the tracking of information along the length of the GI tract, we would recommend placement of multiple synchronized devices in different regions e.g. duodenum, ileum etc., wired to an implanted wireless transmitter.

2) where would be the exit point of the device for the human? Would that be at the back like in the mice or in front? These may have implications if the patients monitoring period is over at least one cycle of wake and sleep since most humans lie on their backs while sleeping.

We thank the reviewer for highlighting this critical consideration for any future development. The primary focus of this manuscript is to establish a foundational platform for recording high-fidelity gut electrophysiology, validated in animal models. For this work, the tethered (wired) approach was instrumental as it provided the high-bandwidth connection necessary to validate the core, sensing technology and capture these complex, multi-frequency signals for the first time *in vivo* in freely-moving subjects. The choice of a dorsal (back) exit point for the percutaneous port in the rat study was primarily for practical reasons related to animal research: it is an area the animal cannot easily reach, preventing damage to the port and wiring. However, the reviewer correctly identifies that a percutaneous lead would present significant challenges for clinical applications. Indeed, for any future translational pathway, the logical progression would be to develop a fully implantable, wireless system. This would involve integrating the sensing array with an internal power and transmission circuit, thereby circumventing the issues associated with a percutaneous exit site altogether. The device would function similarly to a cochlear implant in this sense. While wireless transmission is feasible, it often involves trade-offs in data bandwidth and system complexity, which is why the wired approach was chosen for this initial, validation-focused study.

3) how long do you expect the monitoring to be for humans? in your experiments, I believe you have monitored for a maximum of 3 days. Would your device be able to survive and be still functional within the intended time of monitoring?

In this study, we monitored out to 14 days (see original Fig. 5, now Fig. 6) with little to no loss of function for the device. We anticipate that the monitoring time in a clinical setting would be largely based on the goals of the clinical monitoring scenario. If this device were used in an in-op case, we could presumably examine for recovery of the ENS post-GI manipulation or resection as an indicator for ileus, in which case the device would be placed locally and then removed. In a longer term monitoring scenario, the device would need to remain implanted to

provide meaningful neural data. While our current results indicate feasibility, further studies would be required to evaluate device performance and biocompatibility over extended periods before translating to chronic human use, which is outside of the scope of this study. This incremental developmental approach has been followed in all other implantable technologies that are currently implemented in the clinic.

4) what is the current gold standard of monitoring electrophysiological signals of the ENS? is it possible to just pick one of three situations and also monitor using the gold standard method and compare the findings? all of the experiments are assumed that the leads are measuring what it was intended to measure without a comparative control to ascertain whether the signals picked up by your device is the true signal. If the gold standard method is very difficult, perhaps, you can use your device to measure more easily accessible plexus where both your device and the gold standard method can be used and then compared?

We thank the Reviewer for this important question. As stated in the introduction, currently, there is no gold standard for recording of electrophysiological signals from the enteric nervous system (ENS) *in vivo*. This is the essence of our manuscript – to deliver a new bioelectronic technology capable of acquiring such signals with the high fidelity required for neural recordings; and increase our: (1) limited understanding of the association of gut electrical activity with overall gut function; and achieve (2) limited rigorous electrophysiological analyses required to disentangle the contributions of muscle and neural activity.

Most of what is known about electrophysiological function of the ENS comes from work done *ex vivo* in organ baths. Indeed, we had considered and attempted a strategy involving simultaneous recordings from our engineered device and a conventional *ex vivo* electrophysiological setup by applying a suction electrode for whole nerve recordings of the lumbar splanchnic nerve with a concurrent placement of a flexible device. The details for this setup are noted in the methods section and Extended Data (Extended Data Fig. 5e,f). Specifically, we managed to integrate and synchronize the recording channels between this *ex vivo* electrophysiology rig and the recording setup using our conformal device. However, despite this progress, performing truly simultaneous recordings proved technically challenging for several reasons:

1. The *ex vivo* electrophysiology setup requires the colonic tissue to be mounted in a superfusion bath and continuously perfused both intraluminally and externally with carbogenated Krebs solution. Oils from the interior of the colon that accumulate during the mounting of the tissue in the bath tend to accumulate in the small microelectrode wells of the device, drastically increasing impedance and rendering recordings unreliable or impossible.

2. In the *ex vivo* configuration, the colon is dissected free from adjacent connective tissue, leaving it mechanically unsupported. While this makes it ideal for distension studies and conventional extracellular recordings, it also introduces considerable tissue movement and distortion. Such dynamic changes in the geometry of the tissue make actual implantation of the device into the colon and subsequent positional maintenance very challenging, meaning that a large number of animals would need to be used to acquire data in this fashion. In contrast, the *in vivo* setup allows us to place the device underneath the muscularis externa, where it benefits from the structural rigidity provided by the surrounding tissue. The colon segment remains intact and tethered, preserving the anatomical context, minimizing movement artifacts during distension, and ensuring consistent contact between the electrode array and the target enteric neurons.

While we acknowledge that direct correlation with the gold standard electrophysiology set up in the same preparation would further strengthen our claims, the technical limitations described leave collection of these data beyond the scope of the current work, although we are currently initiating the work required to troubleshoot the quite significant challenges. It should be noted that imaging could be used simultaneously; Calcium imaging has traditionally been used to indirectly measure neuronal activity in the ENS by detecting changes in intracellular calcium concentration, which correlate with action potentials or synaptic activity. This is an important technique that has enabled several key discoveries. However, it has limited applicability to *in vivo*, particularly in freely moving animals, where the large movements of the colon make this technique virtually impossible to apply without the use of highly-invasive mounting of intravital windows, etc. Further, it requires complex protocols with limited translational capacity (e.g., viral injection, transgenic animals) to introduce optical responsiveness, and it lacks the temporal resolution necessary for extracting signals from individual neurons.

Reviewer #1 (Remarks on code availability):

I do not have any understanding of coding. Sorry.

Reviewer #2:

The authors showed the construction and validation of a bioelectronic device to access neural information produced and processed in the gastrointestinal tract. It is interesting. However, one of the major questions is that why you used the flexible electrode rather than stretchable electrode, as we can see the clearly slide between the electrode and the gut in the attached video.

Regarding the use of a flexible/conformable vs. stretchable device, a stretchable device would be very difficult to implant in this scenario. To place the device into the colonic wall, we made a small tunnel and then pulled the flexible device through the tunnel to achieve close contact with the tissue. To avoid issues with mechanical motion, we placed the device into distended portions of the colon, which are already at maximum physiologic strains, thereby ensuring that the conformability of the device could account for the compressive strains experienced during tissue relaxation. The placement in distended tissues also enabled consistent dissection to form the tunnel in which the device was situated. We determined that the surgical placement and technique were unlikely to be compatible with a device that would deform under tension. Additionally, maintaining the device's position in the gut does not require it to expand and shrink, unlike in other implantation sites such as the spinal cord, where stretchability may be necessary to ensure stable contact. For these reasons, we used our flexible/conformable device setup. While we note movement of the device portions that are situated outside the colon wall (Movie 1), we **do not** observe any sliding in our video for the portion of the device that is situated within the gut wall, which contains the recording electrodes. We believe the synchronicity of mechanical motion is most evident in Movie 2. Furthermore, in the context of chronic implantation, the fibrotic response of the tissue is expected to encapsulate the device, helping to secure it in place and further minimizing the risk of displacement over time.

1. In fig.4, you should also give the scale bar of the spike amplitude. In fig.4f, why you only have 20 recordings rather than 28 (in Fig.1, you have 28 recording electrode).

Thank you for noticing that scale bars were missing in Fig. 5 (previously Fig. 4). We have now included them in the figure. The device includes 28 recording channels, as correctly noted. Of these, 20 channels maintained impedance values within our predefined range for high-quality neural recordings. The remaining channels exhibited elevated impedance and were excluded from analysis. This is common practice in electrophysiology, where only channels meeting specific impedance and signal quality criteria are used to ensure data reliability and interpretability. This has been clarified in the manuscript in Fig. 5f (previously Fig. 4f):

"...from the implanted electrodes presenting acceptable (<500 kΩ) electrical impedance, which also presented good signal quality confirmed through offline analysis ..."

2. The pig experiment is also important, why you don't show it as the rat in Fig.4?

Recording stable, spontaneous neural spiking from individual neurons is highly dependent on chronic, unanesthetized preparation. As we note in the manuscript, general anesthesia leads to a "strong suppression of spontaneous neural activity," making the isolation of individual neuron activity inconsistent and exceptionally challenging. The experiments performed in the pig, as well as in the mouse, were acute procedures conducted entirely under anesthesia. The primary goal of these studies was to demonstrate that the device and surgical implantation technique could be successfully scaled to vastly different anatomical sizes, from a small mouse to a large pig, and record physiological responses to stimuli in a dynamic gut environment in the same way that we had shown in anaesthetized rats. Due to the suppressive effects of anesthesia, it was not feasible to reliably resolve putative single-neuron activity in the pig model in the same manner as the chronic rat study.

However, we agree about the importance of the pig experiments and, following your suggestion, have moved this figure from the Extended Data to the main text. This will more clearly and effectively illustrate the platform's

robust and scalable performance, which was the central objective of including these species. This was originally moved to accommodate the figure limit.

3. For the Implant Microfabrication & Integration, you don't mention any encapsulation, or you just forget it?

We refer the Reviewer to our methods section, where we described the encapsulation of the device (Pg. 18):

"Parylene-C was deposited using a dimerized precursor, dichloro-p-cyclophane (Specialty Coating Systems, Indianapolis, IN, USA) onto a Si wafer at 2 μm thickness. Subsequently, wiring was patterned using AZ 5214 photoresist (Microchemicals GmbH, Ulm, Germany) and developed using AZ 351B (Microchemicals GmbH, Ulm, Germany). Gold deposition was performed at 100 nm with a electron beam Evaporator (PVD 75, Kurt J Lesker) using 10 nm of Ti as an adhesion layer to the parylene-C substrate. Gold tracks were patterned using a lift-off technique in acetone. A second parylene-C layer was deposited at 2 μm to insulate gold tracks."

The first 2 μm coat of parylene-C serves as the bottom device, and the second coat of parylene-C provides the top layer of the device. These two layers in total provide the dielectric encapsulation of the device.

4. You should give the device characterization of the electrode, ie EIS.

We thank the reviewer for this valuable suggestion. Following your recommendation, we performed electrochemical impedance spectroscopy (EIS) to characterize the electrodes (Pg. 3):

"...which is visible via electrical impedance spectroscopy (Extended Data Fig. 1)."

Specifically, we measured the impedance magnitude and phase across a frequency range of 1 Hz to 100 kHz on six different electrodes from the same device. The results are now included in the revised manuscript (see Extended Data Fig. 1), where we report the average impedance and phase profiles, along with standard deviations. These measurements confirm the consistency and functional integrity of the electrodes, supporting their suitability for chronic neural recordings.

5. I think the major achievement of this study is the recording of putative single neurons. As the authors mentioned, it is putative and you need to prove it. It is confusing if you want to publish it on Nature Communications when you don't have enough data. And you don't show the pig and mouse data.

As mentioned above, anaesthesia severely dampens ENS neural activity. Both the mouse and pig experiments were acute, under anaesthesia. We considered therefore, that only the rat, chronic data was suitable for resolving putative neuron activity. We subjected these data to a rigorous, multi-step analysis pipeline, standard in the neuroscience community for identifying and validating single-neuron activity from extracellular recordings. As shown in Fig. 5 (previously Fig. 4), a signal was classified as a "putative single neuron" only if it met several strict criteria simultaneously: 1) distinct and characteristic waveform shape typical from neural signals (e.g. asymmetrical); 2) The neuron's firing pattern exhibited a refractory period, meaning it was impossible for it to fire another action potential for a brief period after the first. This is a hallmark of neuronal activity; 3) The spike was largest on one electrode and decayed rapidly on adjacent electrodes, consistent with a localized, point-source

origin (i.e., a single cell body); 4) In principal component analysis, the waveforms from a single putative neuron formed a distinct and stable cluster, separable from noise and other units. Collectively, these criteria provide strong evidence that the signals we identified do indeed arise from enteric neurons. We conscientiously use the term "putative", which is also widely used in brain neuroscience, to reflect the inherent limitations of any extracellular recording where the specific cell cannot be visualized histologically. This is standard practice in the neuroscience community.

This characterization was already described in the manuscript (Pg. 13), copied here with updated reference numbers:

“To investigate putative activity patterns attributable to individual gut neurons, we adapted a thresholding and clustering approach commonly used with brain-derived electrophysiologic recordings^{45,46}. We identified 89 putative single neurons across all recording sessions. These neurons displayed anatomically plausible, localized spatial patterns of action potential waveforms across recording electrodes (Fig. 5f). Importantly, they also exhibited physiological spiking refractory periods (Fig. 5g), distinct patterns of co-activation with other neurons (Fig. 5h), and separable clusters of spatiotemporal action potential features in principal component space (Fig. 5i), supporting the characterization of this activity as representing individual neurons. Spikes from individual neurons were also detected across the time course of the session (Fig. 5j), indicating a stable interface between our device and the tissue.”

But we have included additional explanation in the relevant methodology section (Pg. 24):

“A signal was classified as originating from a "putative single neuron" only if it met several strict criteria simultaneously: 1) distinct and characteristic waveform shape typical from neural signals (e.g. asymmetrical); 2) The neuron's firing pattern exhibited a refractory period, meaning it was impossible for it to fire another action potential for a brief period after the first. This is a hallmark of neuronal activity; 3) The spike was largest on one electrode and decayed rapidly on adjacent electrodes, consistent with a localized, point-source origin (i.e., a single cell body); 4) In principal component analysis, the waveforms from a single putative neuron formed a distinct and stable cluster, separable from noise and other units.”

Reviewer #3:

This is an exciting study from Boys et al. developing a novel technique to record electrophysiological activity in the gut in vivo in free-moving rats. The authors develop a flexible, implantable device and a surgical procedure that enables recording from the enteric nervous system (ENS) while allowing distension and contraction movements of the gut wall. To examine neuronal activity, the authors first performed recordings in vivo under anaesthesia, apply bradykinin and capsaicin to the gut wall during recording, and investigated high vs low frequency responses. In addition, the concentration of isoflurane anaesthesia was investigated. Further, the authors examined activity in free-moving rats, recording activity from what they identify as individual neurons, before, during and after feeding. They also examined the impact of stress on neuronal activity. In addition to rats, the authors have also adapted these implants for larger (pigs) and smaller (mice) animal models.

This is a very exciting and novel advance in this field, enabling researchers to record from the ENS (as well as other electrophysiologically active cells in the gut) in live, free-moving animals. However, there are several major concerns regarding the results.

1. The biggest concern is confirmation of whether the signals that are recorded are arising from enteric neurons. As the authors state, there are multiple types of electrically active cells in the gut wall, including enteric neurons, interstitial cells of Cajal (although they play a smaller role in the colon compared to the stomach), as well as the smooth muscle cells. The device generated by the authors essentially performs extracellular recordings, and while the authors state that they have produced single-unit recordings from individual neurons, the data that is shown does not definitively prove this. Can the authors provide more evidence that the recordings that they identify as “neuronal” do indeed arise from neurons?

A vital experiment would be to perform recordings using this device coupled with simultaneous recording from enteric neurons (either intracellular recording or calcium imaging), in an ex vivo setting. This has previously been

achieved using a somewhat reliable recording device (Athavale et al., 2025 PMID 38860855). The authors also use high and low bandpass filters to specifically show high and low frequency activity. It is unclear why they did this, and how this relates to the identification of neuronal activity. Acknowledging my lack of familiarity on the engineering aspect of this, some explanation and/or references on this filtering and signal identification would be helpful to the neurogastroenterology audience.

The lack of clarity on this vital point shadows the various in vivo experiments that were performed, such as stress and feeding-associated changes in activity. If we cannot be sure that these responses do indeed come from enteric neurons, it becomes difficult to extrapolate and interpret the data from these experiments.

The reviewer's primary concern is the definitive identification of recorded signals as "neuronal." It is possible that the use of the term putative undermines the reviewer's confidence in our data. However, we conscientiously use the term "putative", which is also widely used in brain neuroscience, to reflect the inherent limitations of any extracellular recording where the specific cell cannot be visualized histologically. Our approach to understanding the nature of the signals recorded by our devices is three-fold: 1) electrode placement and tissue properties, 2) frequency-based filtering to separate fast neural spikes from other slower signals, as widely conducted in neural electrophysiology studies, and 3) rigorous spike-sorting analysis based on established neurophysiological principles, such as extracellular waveforms of action potentials and physiological refraction.

1. Our device is oriented with the dielectric backing facing the myenteric plexus and smooth muscle layers, which limits the contribution of high-frequency signals from these deeper regions. This orientation, combined with the insulating properties of the dielectric layer (parlylene-C), is designed to attenuate high-frequency components from distant sources, as supported by prior studies on signal propagation and insulation in electrophysiological recordings (Lee et al. 2025, [10.1016/j.sna.2025.116437](https://doi.org/10.1016/j.sna.2025.116437)). However, it is important to note that tissue inherently possesses low-pass filtering properties (Gygi & Moschytz, IEEE, 1997), meaning that low-frequency signals, such as those arising from smooth muscle or myenteric ICCs, can propagate further and still be detected by extracellular electrodes, even if the source is not in direct proximity. This principle is well established in brain recordings (e.g., EEG) and in the gut via electrogastragrams (EGG), where slow-wave activity is recorded from the skin surface. In contrast, high-frequency signals (e.g., neuronal spikes) require close proximity to the source, typically within tens of microns for PEDOT:PSS-coated electrodes. As the electrical source is moved farther away relative to the electrode placement, the amount of high-frequency signal that contributes to the overall signal from that particular source decreases. In our case, the presence of high-frequency signals strongly suggests that the device is in close proximity to active neurons within the submucosal plexus, which would be unresolvable if the distances from the device to source were greater. The low-frequency components, however, may reflect broader network activity, including contributions from smooth muscle or ICC, even if these are not directly adjacent to the electrode. The device, by its nature as an extracellular recorder, captures the sum of all local electrical events, including those at high-frequency, as well as more distant low-frequency events.

2. In the experiments conducted under anesthesia (e.g., mechanical distension and pharmacological stimulation), the primary goal of filtering is to separate the fast, transient signals from neurons from the slower, more sustained responses of other cells like smooth muscle. Neuronal signals are extremely rapid events, lasting only few milliseconds. This speed translates to high-frequency content in the recorded electrical signal. By applying a high-pass filter that only allows frequencies above 300 Hz to pass through, it is possible to isolate these brief neural spikes from the slower background signals. For example, during mechanical distension experiments, an *"initial fast peak in the high-frequency trace"* was observed, which is indicative of a rapid neural response. In contrast, the electrical activity of smooth muscle cells and ICCs is much slower. A low-pass filter that only allows frequencies below 300 Hz to pass through captures these sustained, low-frequency voltage responses. This allowed us to observe an *"asynchronous extended voltage response, visible in the low-frequency trace"* following the initial neural spike during distension, suggesting the subsequent action of other, slower cell types like myenteric or submucosal ICCs, which have also been identified in the rat colon (e.g., Lee et al., J Neurogastroenterol Motil 2017). We acknowledge that this was not sufficiently explained in the manuscript and has amended it accordingly and included this new reference (Pg. 5):

"We examined the electrophysiological response at both high frequency (300-2000 Hz), to isolate the neural components from the signal, and low frequency (0-300 Hz), to isolate the slower components of the signals

arising from other cells, like submucosal and myenteric ICCs and smooth muscle cells in 10 s windows after each distension (Fig. 2c).”

and (Pg. 6):

" This two-component signal, featuring an initial fast peak in the high-frequency trace followed by a slower, extended low-frequency response, suggests the involvement of multiple cell types. The high-frequency activity reflects neuronal firing, consistent with the primary role of the ENS in initiating the colonic motor response, and confirms close proximity of the electrodes to submucosal neurons. High-frequency signals from more distant sources, such as the myenteric plexus or muscle layers, are likely attenuated due to the low-pass filtering properties of biological tissue and the dielectric backing facing the myenteric plexus and smooth muscle layers³⁰. In contrast, the slower low-frequency component likely arises from the integrated activity of the broader neuromuscular network. Although the device is oriented away from the myenteric plexus, low-frequency signals from more distant sources, including smooth muscle and ICCs in both the submucosal³¹ and myenteric layers, which are electrically active as demonstrated in gastric motility network ³², can still propagate through tissue and be detected. We note that while colonic motility is predominantly neurally mediated, there is evidence that myogenic mechanisms can contribute under certain conditions³³. We leveraged this physiological feature to validate device placement and recording capabilities via distension-evoked responses.”

In freely-moving animals, we used a frequency decomposition approach similar to what is done in EEG and ECoG analyses to parse the complex signals into distinct bands, each associated with a different biological source. This methodology, with its corresponding references, is detailed in the "Stress-response multifrequency band analysis" section of the Methods and in the caption in Figure 5. As an overview: [0 - 0.2 Hz] band is used to capture the very low-frequency "slow wave" activity that is characteristic of the gut's pacemaker cells, the Interstitial Cells of Cajal (ICCs), originated locally in the submucosal plexus or potential volume-conducted components from deeper layers such as the myenteric plexus; [0.2 - 1 Hz] & [1 - 5 Hz] bands are designated for the slow and faster rhythmic activities generated by the gut's smooth muscle layers, which are responsible for contractions and motility; [5 - 300 Hz] band primarily captures electromyographic (EMG) signals related to the broader movements of the animal's body and tissue, involving skeletal muscle activity; [300 - 2000 Hz] band is used to isolate the fast spiking activity of neurons within the ENS.

3. By filtering for only these high frequencies, we could then confidently proceed to the spike-sorting analysis, standard in the neuroscience community, for identifying and validating single-neuron activity from extracellular recordings. As shown in Figure 4, a signal was classified as a "putative single neuron" only if it met several strict criteria simultaneously:

1. Distinct and characteristic waveform shape typical from neural signals (e.g. asymmetrical);
2. The neuron's firing pattern exhibited a refractory period, meaning it was impossible for it to fire another action potential for a brief period after the first. This is a hallmark of neuronal activity;
3. The spike was largest on one electrode and decayed rapidly on adjacent electrodes, consistent with a localized, point-source origin (i.e., a single cell body);
4. In principal component analysis, the waveforms from a single putative neuron formed a distinct and stable cluster, separable from noise and other units.

Collectively, these criteria provide strong evidence that the signals we identified do indeed arise from enteric neurons.

Regarding the additional experiments in an *ex vivo* setting, we refer to our response to Comment 4 by Reviewer 1.

In summary, our approach supports the neuronal origin of the recorded activity. We hope this clarification addresses the reviewer's concerns and provides a better understanding of the rationale behind our methodological choices.

2. The authors state that they are recording from the submucous plexus of the ENS. They state that the device is placed “underneath” the muscularis external (page 2), and the recording electrodes directed lumenally.

However, from the figures shown, it's difficult to confirm this. From Fig 1 and Supp fig 1, the location of the "tunnel used for implant placement" is shown. However, depending on the orientation of the tissue when it was sectioned (transverse or longitudinal), it is extremely difficult to identify which cell layers are around this tunnel. Presumably, the device was removed before tissue processing and histological analysis, hence its precise location is difficult to ascertain. As a first point, some higher magnification H&E images would be helpful. In addition, some immunohistochemical (or immunofluorescent) staining of the tissue to distinguish between the different cell types and also identify the location of enteric neurons is needed. A further point is that annotation of the images to help orient the reader would be beneficial for many of the figures (please see specific comments below).

We thank the review for this important point, and we have added further explanation to detail the data we have collected. The histological images taken show a cross-section of the gut, where the tunnel and subsequent implant are oriented axially, oral-to-anal. We have added further images to the extended data showing the placement, and we have also included annotation of these images as well in Extended Data Fig. 3.

While an additional immunohistological study would indeed be interesting, the correlation of these data with the region of the gut where the electrodes collect data would not be possible. The implanted device contains 28 recording electrodes, arranged into 7 sets of 4 electrodes, spaced axially along the colon. The recording pads themselves are 20 μm in diameter, and the underlying wiring is 100 nm thick. This means that imaging the electronics themselves histologically is physically impossible using optical techniques. So, while we could in theory identify that the device is near neurons, we could not identify which neurons are near which electrodes. Further, as the gut contains multiple neural types, determining which neurons are firing at a given time would also not be possible using such an endpoint analysis.

Lastly, throughout the manuscript, the authors refer to recording electrical activity that is more related to the function of myenteric neurons, and myenteric and muscular ICC activity. Therefore, it is key to identify whether the device is indeed recording from submucous plexus activity, or the external muscle (plus myenteric) layers.

We thank the reviewer for this observation. We believe this point has already been addressed in our response to Comment 1, where we clarify the device orientation and how signal frequency content supports recording from submucosal neurons rather than from other layers.

3. To strengthen the study and its implications, the authors may find it beneficial to collaborate with researchers or experts who have extensive experience and specialised knowledge in the enteric nervous system and gastrointestinal physiology. There are several instances throughout the manuscript where more specificity with wording and references, as well as more detailed information on the anatomy and neurophysiology is needed (please see below for specific comments).

We thank the reviewer for the helpful suggestion. We wish to clarify that the team includes strong expertise in both enteric nervous system (ENS) and gastrointestinal (GI) physiology. Professor Niall P. Hyland, based at University College Cork and a co-author on this study, received specialist postdoctoral training in ENS electrophysiology at the University of Calgary, a leading center in the field. His subsequent career has focused on gut physiology and electrophysiology, gut-brain signaling, visceral sensitivity, and microbiota-host interactions. Dr. Friederike Uhlig, also a co-author, contributes further expertise in GI physiology, particularly through her work demonstrating that bacterial products can directly modulate sensory neurons and alter gut motility and secretion independently of immune signalling (iScience, 2020). While the use of implantable bioelectronics is a novel development, it is well supported by Professor Hyland's extensive expertise in the relevant physiological systems, along with the complementary strengths of the wider team.

4. More thorough proof-reading of the manuscript would be helpful. There are many small errors in this manuscript, although individually these are all very minor, the cumulative effect is very distracting. For example, it would have been helpful to see greater consistency around the use of acronyms, and also greater consistency around formatting - some words are italicised while others are not. The use of line and page numbers would have also helped with the review process.

We thank the reviewer for this helpful observation. In response, we have conducted a comprehensive proof-reading of the manuscript to correct all typographical and formatting errors. We also note that our initial

submission follows Nat Comms submission template, which does not contain page numbers. We have now included page numbering to facilitate the review.

Specific comments by Section:

Summary:

5. Sentence 4: The authors use the phrase “the constant intrinsic and extrinsic motion” of the gut. What do they mean by extrinsic motion? There is extrinsic innervation of the gut, but generally, there is not much reference to “extrinsic motion” of the gut.

We use extrinsic in this case to refer to its standard definition, “originating from the outside.” We chose this wording to differentiate the overall motion of the gut within the peritoneal cavity, relative to the other organs, from the peristaltic motion of the gut, as actuated by smooth muscle. However, to avoid confusion, we have changed the wording (Pg. 2):

“However, the constant motion of the gut, arising from its relative movements in the peritoneal cavity, in addition to the peristaltic movements, associated with gut motility, as well as the sparse distribution of the neurons that constitute the enteric nervous system, has made access and analysis for study of this important component of the gastrointestinal tract exceedingly challenging.”

6. Final sentence: ENS has already been used as an acronym already used prior to this point

We appreciate the reviewer’s attention to clarity. However, we have not introduced any acronyms prior to this point in the manuscript, including “ENS.” Therefore, we are unsure which earlier usage the reviewer is referring to. We would be happy to revise the text if further clarification is provided.

Introduction:

7. Paragraph 2, sentence 1: the authors state that the ENS plexi “wrap circumferentially around the GI tract”, this phrasing potentially discounts the continuity of the plexi in the longitudinal direction. The ENS in healthy conditions spans the entire length of the gut as well as spanning the circumference.

We have adjusted the wording to reflect this (Pg. 2):

“The ENS is organized in two interconnected ganglionated plexi (myenteric and submucosal) that run along the length of and wrap circumferentially around the GI tract¹, with the former plexus situated between the outer muscle layers and the latter plexus in the submucosa.”

8. Paragraph 2: the authors state that past recordings of the ENS have focused on *ex vivo* recordings with “patched electrophysiological setups”, presumably they mean patch-clamp recordings from *ex vivo* preparations of the gut. However, this is not strictly accurate as there are very few studies that have used whole-cell patch-clamp to investigate the activity of enteric neurons. Most studies have used sharp electrode intracellular recording. While this may seem like a small point of difference, it highlights a lack of familiarity with the literature in this field.

We thank the reviewer for their careful reading and for this important correction. This distinction is important for accurately representing the state of the literature. We have amended the manuscript and included more references to previous studies to correct this lack of precision (Pg. 3):

“Consequently, previous studies have predominantly relied on *ex vivo* recordings in organ baths, employing intracellular techniques, most commonly using sharp electrodes⁹ or voltage-sensitive dye (VSD) imaging¹⁰. Calcium imaging has also been used *in vivo* under anesthesia to indirectly assess neural activity and synaptic dynamics^{11,12}.”

9. Paragraph 2, last sentence: The authors state that there are “no current devices demonstrating successful in

vivo recording of ENS electrophysiology". However, there are some studies that have achieved this, for example <https://doi.org/10.1152/ajpgi.00069.2024> uses flexible multielectrode arrays to record electrical signals in the rodent stomach in-vivo. There is also evidence of recordings from the porcine stomach PMID: 28035728 and 35726361, and in human stomach <https://doi.org/10.1152/ajpgi.00125.2010>. The animal studies have been performed under anaesthesia, therefore the current work is highly novel and a great advance to be able to perform these recordings in conscious, free-moving animals. However, recognition of the previous studies should also be taken into account.

We agree that our statement, "*no current devices demonstrating successful in vivo recording of ENS electrophysiology*", may be seen as an oversimplification of the current state of the field, and we are taking the opportunity to acknowledge these studies and better contextualize our work. Our intention was to highlight the specific challenge of recording high-frequency, putative single-neuron activity from the enteric nervous system, particularly in the distal gut of conscious, freely-moving animals. The studies that the reviewer has pointed out are indeed significant contributions to *in vivo* gastrointestinal electrophysiology from the stomach and for slow-wave recordings, and we have included (references #18-21) them together with other relevant studies (references #9-12) as noted in response to Comment 8, in a revised introduction that acknowledges the prior art while clarifying our specific advance (Pg. 3):

"Even with these advancements, few studies to date have been able to access intestinal tissue *in situ* using *in vivo* implantable technologies^{16,17}, and few have also been able to capture gastric electrical activity, such as slow waves, in anesthetized animals and humans¹⁸⁻²¹. However, recording high-resolution, putative single-neuron activity from the ENS in the gut of freely moving animals remains an unmet technical and biological challenge."

Implant Design & Surgery Development

10. Figure 1d- 1i: These panels would benefit with some more labelling to orientate the reader. Panel d-i, labelling of the small intestine vs colon would be helpful, as well as other structures that are visible. Also why are the surgical images in this figure so pink? It seems the contrast has been altered significantly

This image shows only a distended section of colon. No small intestine is visible. Some visceral adipose tissue is visible adjacent to the colon. Otherwise, the only visible tissue is subcutaneous musculature and the cross-section of the peritoneal wall. The surgical images are not altered. We took these images through the surgical stereoscope during the surgery. We have added to a figure to the Extended Data labelling the adipose tissue (Extended Data Fig. 2).

11. Figure 1 legend: There is reference to a red box when discussing the loops designed for suturing the device however there is a red box around panel j which has no relevance to this. Please check what you are trying to refer to in the figure and update the legend accordingly.

We apologize for this oversight. This is an iteration from a past version of the manuscript. The text should indicate this as a "black box," which has now been adjusted.

12. Figure 1j: additional landmarks need to be added to this panel, it is incredibly difficult to orientate the key features. It does seem like the cavity where the electrode was is between some muscle layer and the muscularis mucosa which would indicate the submucosal space however if the recording interface was facing internal there is no evidence in this image of any neurons being present in that space. The authors might look to a better representative image being used here. What anatomical plane were the sections taken? Transverse across the full circumference of the tube?

These slices are a cross-section of the colon wall, taken at a slight angle to accommodate the placement of the implant into the colon. As noted in the text, these slides were stained with hematoxylin and eosin. As such, specific staining of neurons would not be expected. We have added to the Extended Data with more complete annotation of this image (Extended Data Fig. 3).

Distension of gut wall validates physiological responses to mechanical stimuli:

13. Paragraph 1, first sentence: the authors describe a lot of physiology in this first sentence, but cite a reference that mostly focuses on the neurochemical subtypes of enteric neurons. There are many other more physiologically relevant studies and reviews that could be used here, including PMID 32152479 (which is already cited).

Thank you for this suggestion, we have now included two more citations, one to Spencer, et al 2020, PMID 32152479 (already cited in the manuscript, now under reference #26) and another to this relevant review (Xue et al. 2007, DOI: 10.1016/j.autneu.2007.02.001). We have kept citation to Furness et al, 1980, which we believe is still relevant (now under reference #25):

“During the passage of fecal matter through the colon, the gut wall experiences substantial strain, which activates stretch-sensitive neurons and drives a cascading peristaltic response ^{25–27}”.

14. Supplementary video 3: The tissue looks very dry in this video. Were the authors able to maintain tissue integrity through these experiments?

Yes, saline was added topically to the colon periodically, approximately once per minute throughout the surgeries to maintain tissue hydration. For imaging purposes, the tissue was patted dry to avoid reflections and optical artifacts caused by surface fluid, which would otherwise obscure structural details in the image.

15. Paragraph 2: the authors state that in response to mechanical distension of the gut, they see “asynchronicity of the response” which implies that multiple cell types are responsible for the mechanical distension. And further suggest that ICCs which are “crucial intermediates in the GI tract for bridging neurons to muscle tissue” have contributed to the recorded activity following distension. There are many issues with this.

Firstly, colonic motility is mostly neutrally mediated, not myogenically mediated. Indeed, ICC are interconnected within this network but they are not essential while the neuronal network is. Therefore, the dominant signal here should be from the neurons. While there may be a slow wave component that involves ICCs, these are not as prominent in the colon as they are in the stomach. Therefore, the authors should presumably expect to see a dominant frequency of activity from neurons, instead of a series of asynchronous spikes? The citation of reference #25 has a strong focus on gastric motility networks, which is very different from that of the colon. For the colon the critical component of this network for motility is the ENS not the ICC.

Secondly, the authors previously state that they have oriented the recording device so that activity from the myenteric plexus and smooth muscle layers, and so presumably myenteric and muscle associated ICCs are not being recorded. However, this contradicts the current statements that signals from ICCs are contributing to the distension-evoked responses. Given the placement of the electrodes interfacing with the submucosal plexus where there are less neurons and less ICC, there is potential that single spikes are that would be detected. An additional complication to this is that submucosal neurons in rodents are generally less involved in control of gut motility. While they may respond to distension-evoked activity, it is uncertain what is the importance of detecting their signals with this mechanical stimuli.

Thank you for these insightful comments. We fully agree with the reviewer’s assessment that colonic motility is primarily driven by the ENS, and that neurons are the essential component of this network. Our observation of an “initial fast peak in the high-frequency trace (300-2000 Hz)” following distension is indeed consistent with this principle, representing the expected dominant and rapid neuronal response that initiates the motor pattern. Our mention of “asynchronicity” and the potential contribution of multiple cell types, including ICCs, was intended to explain the subsequent, more complex part of the signal, specifically the “asynchronous extended voltage response, visible in the low-frequency trace (0-300 Hz)”. We hypothesize that after the initial, synchronized neural firing, that this slower component reflects the integrated activity of the broader neuromuscular network, including smooth muscle and possibly ICCs, which are electrically active and coupled to both neurons and muscle.

We acknowledge that ICCs are not the primary drivers of colonic motility and are less prominent in the colon than in the stomach. However, subtypes such as submucosal ICCs (ICC-SMP) have been identified in the rat colon (e.g., Lee et al., J Neurogastroenterol Motil 2017) as described in response to Comment 1, and depending on the precise positioning of the implant, it is possible that our device detects signals from these cells, particularly in the low-frequency range. Previous reference #25 (Sanders, et al., Annu Rev Physiol, 2006, now reference #34) was cited to support the general electrical activity of ICCs, though we acknowledge its primary focus is on gastric networks and have clarified this in the revised text (Pg .6):

“Although the device is oriented away from the myenteric plexus, low-frequency signals from more distant sources, including smooth muscle and ICCs in both the submucosal³³ and myenteric layers, which are electrically active as demonstrated in gastric motility network ³⁴, can still propagate through tissue and be detected.”

We agree that our statement regarding the orientation of the device in the submucosal plexus facing towards the lumen, may appear to contradict the suggestion that ICCs contribute to the recorded signal. We refer here to our response to Comment 1 on this regard. Briefly, low-frequency signals propagate farther through tissue, as evidenced by dermal recordings of EMG, EGG, and EEG recordings.

We also note that while colonic motility is predominantly neurally-mediated, there is evidence that myogenic mechanisms can contribute under certain conditions. For example, in the presence of tetrodotoxin (TTx), which blocks neuronal activity, colonic tissue can still respond to pharmacological stimulation such as acetylcholine analogues (e.g., DOI: 10.1152/ajpgi.00227.2013). This suggests that in our preparations, compounds like capsaicin and bradykinin may elicit responses through both ENS and direct muscle activation. After all, the submucosal and myenteric plexus do not operate in isolation, and there is communication between both. We leveraged this physiological feature to validate device placement and capabilities via distension-evoked responses.

As noted in Comment 1, we have amended the original wording to clarify the technical and biophysical basis for our interpretation (Pg. 6):

"This two-component signal, featuring an initial fast peak in the high-frequency trace followed by a slower, extended low-frequency response, suggests the involvement of multiple cell types. The high-frequency activity reflects neuronal firing, consistent with the primary role of the ENS in initiating the colonic motor response, and confirms close proximity of the electrodes to submucosal neurons. High-frequency signals from more distant sources, such as the myenteric plexus or muscle layers, are likely attenuated due to the low-pass filtering properties of biological tissue and the dielectric backing facing the myenteric plexus and smooth muscle layers³². In contrast, the slower low-frequency component likely arises from the integrated activity of the broader neuromuscular network. Although the device is oriented away from the myenteric plexus, low-frequency signals from more distant sources, including smooth muscle and ICCs in both the submucosal³³ and myenteric layers, which are electrically active as demonstrated in the gastric motility network ³⁴, can still propagate through tissue and be detected. We note that while colonic motility is predominantly neurally mediated, there is evidence that myogenic mechanisms can contribute under certain conditions³⁵. We leveraged this physiological feature to validate device placement and recording capabilities via distension-evoked responses.”

16. Figure 2: Additional labels would help orient the readers

Labels have been added.

17. What are the benefits of the high and low bandpass?

Please, refer to our answer provided in response to the Reviewer's Comment 1.

Pharmacological stimuli initiate multi-frequency electrophysiological response with common neurophysiological characteristics

18. Why was bradykinin and capsaicin topically administered? In particular, this is not a relevant stimulus for the submucosal plexus. We recognise that technically this was a more straight-forward route of administration,

however, intraluminal administration (which was provided in the case of capsaicin) would have been more physiologically relevant.

We agree with the reviewer that luminal administration might preferentially reach the SMP. However, in our previously published studies (Peiris et al., 2017, DOI: 10.1177/1744806917709371; Barket et al., 2022, DOI: 10.1113/JP283170), superfusion of bradykinin and capsaicin onto the serosal surface reliably elicited robust neuronal discharge from colonic neurons. These findings validated topical application as an effective and reproducible method to activate enteric neurons and guided our approach in the current study. In fact, our choice was driven by both technical and ethical considerations. Intraluminal administration poses a significant limitation; it is difficult to ensure complete clearance of luminal stimulants, which would have required terminal experiments for each dosing condition. This would have significantly increased the number of animals used. Instead, we opted for topical administration to test multiple compounds within the same preparation, allowing us to assess the feasibility and reproducibility of electrophysiological responses (as shown in Fig. 3). This approach was consistent with the principles of the 3Rs, specifically Reduction, by minimizing the number of animals needed while still achieving our scientific objectives.

19. Could the authors also provide some more information on the topical administration route, ie. was there any shielding of extrinsic neural pathways within the abdominal cavity that may influence these signals that are being identified?

We agree with the Reviewer that, given the constraints imposed by the *in vivo* setup, we cannot rule out that extrinsic signals were detected. However, we would like to refer the Reviewer to our response to Comments 1 and 15 on our approach to identify the signal's origins and the explanation of the low-pass filtering properties of tissue, which degrades signals that are not locally originated.

20. Could post-hoc analysis of the plexus neurons in the implant region be completed to identify the chemotype/phenotype of the neurons being recorded from?

We refer the reviewer to our response to Comment 2. Briefly, while this would be very interesting, it is technically infeasible to determine the relative positioning of the implant recording sites to local neurons.

21. Do the authors have any immunohistochemical staining of the plexi/neurons post implantation to investigate damage/integrity?

Please see our response to Comment 2. Briefly, we did not perform post-analysis immunohistochemical staining, as we would not be able to isolate the positioning of the device relative to local neurons.

Conformable devices enable to record colonic responses to feeding and stress

22. Why is it necessary to filter to capture slow waves from the submucosal plexus? There will be no slow waves generated from ICC-submucosa this is a ICC-myenteric and ICC-intramuscular phenomenon. Please see point above. While it seems across this manuscript their aim is to record from the submucosal ganglia not the myenteric ganglia but there are references to activities that are controlled primarily from the myenteric ganglia when these neurons are not the ones being recorded from?

We agree that classical slow waves are generated by ICCs located in the myenteric plexus and intramuscular layers. However, as noted in our response to Comments 1 and 15, we have cited evidence suggesting the presence of ICCs in the submucosal plexus, which, although less prominent, may contribute to local electrical activity under certain conditions. Furthermore, as discussed, biological tissue exhibits low-pass filtering properties, allowing low-frequency signals to propagate over greater distances and be detected even when their sources are not in direct proximity to the recording electrodes.

Accordingly, our intention in defining the frequency bands was not to suggest that classical slow waves originate from the submucosal plexus, but rather to justify the inclusion of low-frequency bands in our signal processing strategy. Specifically, the 0–0.2 Hz band was included to capture potential ICC-related slow components that

may arise either locally from submucosal ICCs or via volume conduction from deeper layers such as the myenteric plexus or smooth muscle.

We have amended the wording in the caption of Fig. 6 (previously Fig. 5) and the methods section to clarify this:

“The frequency bands used were: "0 - 0.2 Hz" to capture slow wave activity associated with interstitial cells of Cajal (ICCs)⁵³, originated locally in the submucosal plexus or potential volume-conducted components from deeper layers such as the myenteric plexus; "0.2 - 1 Hz" for slow rhythms such as those in circular smooth muscle^{54,55}; "1 - 5 Hz" for faster rhythmic activities in smooth muscle^{54,55}; "5 - 300 Hz" for primary skeletal muscle activity related to tissue and body movements (EMG related)⁵⁶; and "300 - 2000 Hz" for high-frequency neural components^{48,57–59}.”

Conclusion

23. Final sentence: while we appreciate that this newly designed flexible array provides a great advance in terms of being able to record from free-moving, in vivo animals, recognition of previous work where recordings were performed in vivo is also important. Greater familiarity with these references can perhaps help the authors interpret their signals and also hone in on the question of which responses correlate directly with ENS activity.

We sincerely thank the reviewer for their thorough and constructive feedback across all comments. We have carefully addressed each point and revised the manuscript accordingly to improve clarity, accuracy, and contextual interpretation of our findings. In particular, we have clarified the anatomical and physiological basis of our recordings, acknowledged relevant prior in vivo work and other relevant literature, and refined our discussion of signal origin and interpretation. We hope that the changes made in response to these comments now fully address the reviewer's concerns and enhance the overall quality and rigor of the manuscript.

Methods

Device Design

24. Sentence 2: the inter-ganglionic distance mentioned here and in the referenced paper is for myenteric plexus not the submucosal plexus. The submucosal plexus is sparser than the myenteric and would have larger inter-ganglionic spaces

We thank the reviewer for highlighting this important anatomical distinction. We agree that the inter-ganglionic spacing of ~250 μm refers specifically to the myenteric plexus, where ganglia are more densely and regularly distributed. This detail is important and has now been clarified in the manuscript. While the submucosal plexus is more sparsely populated, we were unable to find a published value for inter-ganglionic spacing in rats. However, available data from other species suggest that the spacing remains within the same order of magnitude. We therefore consider our 250 μm electrode spacing appropriate for targeting both plexuses, particularly given the spatial reach of volume-conducted signals (Pg. 17):

“The inter-ganglionic spacing in the myenteric plexus for proportions of the ENS in a rat is approximately 250 μm ⁶⁰. The submucosal plexus is more sparsely populated, resulting in larger distances between ganglia, but still within the same order of magnitude. This supports the relevance of our electrode design for capturing activity across both plexuses.”

25. Sentence 11-12: It is stated that the arrangement of the microelectrodes would "theoretically enable the identification of single units. However, we did not utilize this capability in this study." But the authors state in the paper that they obtained single neuron data. Are these different things?

We apologize for this oversight. This is an iteration from a past version of the manuscript. We have removed that wording to avoid confusion.

26. Again, in the final sentence the ENS acronym has already been used.

That wording has been removed altogether when addressing Comment 25.

Implant Microfabrication & Integration

27. Sentence 3: grammatical error *an*

Corrected.

Drug preparation and administration

28. Sentence 3: grammatical error of instead of on

Corrected.

Analysis of drug stimulation trials

29. Re-word sentence 5 grammatical error

Corrected.

Chronic surgery & electrophysiology

30. Last sentence: need to reword this sentence

We have reworded the last sentence (Pg. 23):

“Recordings were conducted on Day 1, 8, and 12.” with: “Electrophysiological recordings were obtained on Days 1, 8, and 12 following device implantation.”

Supplementary

Figure 2

31. The figure legend says urethane, but the data shows only high and low isoflurane?

Fig 4G and H indicate the urethane recordings, as is noted in the caption. We have added a label in the figure to clarify it further.

Figure 6

32. The data in the figure shows mouse, rat and pig but the title only includes mouse and pig

Corrected.

Reviewer #3 (Remarks on code availability):

n/a

Reviewer #4 (Remarks to the Author):

Response to reviewers

We would like to thank all the reviewers for their valuable comments and suggestions.

REV 1

The authors have sufficiently addressed all of my concerns. It is an impressive work and I believe the field would benefit greatly from this work. Congratulations, authors.

Thanks are due to the reviewer for the time and care they took to help improving our manuscript.

REV 2

1. It's not a good answer regarding the use of a flexible vs. stretchable device. If you can place the flexible device into the colonic wall, you can do the same for a stretchable device.

We respectfully disagree with this point. The placement of a stretchable device for this application is notably more difficult given the tissue structure. This specifically relates to the thinness of the gut tissue and the difficulty in driving a device into these inter-tissue space.

2. As the author mentioned that the device includes 28 recording channels, as correctly noted. Of these, 20 channels maintained impedance values within our predefined range for high-quality neural recordings. The remaining channels exhibited elevated impedance and were excluded from analysis. Please think about why? The reason is that the clearly slide between the electrode and the gut. Then the 8/28 electrodes cannot contact with the colonic wall, which resulted in high impedance.

We disagree with the reviewer that this is about poor contact. We fabricate devices in an academic cleanroom, where the yield is not the same as in a commercial cleanroom. We undertake quality control post implantation, to control also for problems with electrodes post implantation. We apply thresholds and exclude electrodes with high impedance which is standard practice. High impedance results from either too much insulating or not enough conducting material. A genuine tissue–electrode interface issue, e.g. sliding, would manifest as fluctuations in impedance over time reflecting variable contact, rather than as persistently high impedance values.

3. Regarding the general anesthesia leads to a "strong suppression of spontaneous neural activity," it is easy to reduce the anesthesia and then you can get a good signal. The reviewer trivialises this point which is certainly not "easy". The effects of anaesthesia on gut neural activity are well known to gastric and general surgeons. The post-operative condition, ileus, specifically relates to this factor, which occurs commonly during surgical anaesthesia. Significant optimisation was required to investigate the correct anaesthetic to use on rats. We point the reviewer to Figure 2 where we did indeed reduce the anaesthesia and successfully record neural activity. Lowering anaesthesia further would have posed serious animal welfare risks, including the potential for the animals to regain consciousness, which would violate ethical and licensing regulations. Such experiments were not allowed to be carried out on pigs.

4. For the Implant Microfabrication & Integration, you don't mention any encapsulation. What I mean is that how you encapsulate the flexible electrode with the cable or connector. What you talked about is Microfabrication, not Integration.

We apologise for any misinterpretation. The implants are bonded to a flat, flexible cable (FFC) using anisotropic conductive film, bonded via a flip-chip bonder. Anisotropic conductive film contains

conductive microbeads in an insulating matrix, isolating the tracks from each other and confounding shorts. This site is completely sealed to water leakage without the need for additional encapsulation, as we have consistently verified in all our projects and applications. The FFC was then externalised through a subcutaneous port, which allowed us to connect to the acquisition system. Please see the following section from the text (pg. 16):

“Finally, devices were removed from the Si wafer for bonding and integration with external electronic components. Implants were bonded (FINEPLACER pico2, Finetech GmbH, Germany) using an anisotropic conductive adhesive film (ACF) (10 µm particle size, 3T-TGP20500N, 3T Frontiers, Singapore) to flexible flat cables (FFCs) using either commercial FFCs (Molex, Lisle, IL, USA) or a custom FFC (Fig. 5a) that we designed and produced for chronic measurements.”

5. for the recording of putative single neurons, there are a lot of parameters that can be used to character the “spike”. Please ref Moore et al., Science 355, eaaj1497 (2017).

We have studied the article in question. We note that this is not the original work on cortical spike characterisation. We chose to reference the original work as being more appropriate, our reference 49.

Sirota, A. *et al.* Entrainment of Neocortical Neurons and Gamma Oscillations by the Hippocampal Theta Rhythm. *Neuron* **60**, 683–697 (2008).

<https://doi.org/10.1016/j.neuron.2008.09.014>

REV 3

The current device is recording from the submucous plexus, and we appreciate how the authors have carefully addressed that while most electrical signals from the myenteric plexus has been attenuated, some information from the myenteric neurons, ICCs and outer smooth muscle cells can still be detected, and also, information is likely to be relayed to via submucous neurons. Given the importance of the myenteric plexus in the control of gut motility, how easily can this device be adapted to record from the myenteric plexus instead of the submucous plexus? Could the authors make a comment on this?

We thank the reviewer for this insightful comment. As described, in our current approach, the device is placed beneath the muscularis externa, where the surrounding tissue provides structural support and orients the electrodes toward the submucosal plexus. In principle, the same device could be reoriented to target the myenteric plexus by directing the recording sites toward the intermuscular plane. This would likely require implantation within the muscularis externa itself, necessitating careful optimisation of the surgical procedure to avoid disrupting the muscle tissue.